

# Dissipation-induced topological insulators:
# A no-go theorem and a recipe

**Moshe Goldstein**⋆

Raymond and Beverly Sackler School of Physics and Astronomy,
Tel Aviv University, Tel Aviv 6997801, Israel

⋆ mgoldstein@tauex.tau.ac.il

## Abstract

Nonequilibrium conditions are traditionally seen as detrimental to the appearance of quantum-coherent many-body phenomena, and much effort is often devoted to their elimination. Recently this approach has changed: It has been realized that driven-dissipative dynamics could be used as a resource. By proper engineering of the reservoirs and their couplings to a system, one may drive the system towards desired quantum-correlated steady states, even in the absence of internal Hamiltonian dynamics. An intriguing category of equilibrium many-particle phases are those which are distinguished by topology rather than by symmetry. A natural question thus arises: which of these topological states can be achieved as the result of dissipative Lindblad-type (Markovian) evolution? Beside its fundamental importance, it may offer novel routes to the realization of topologically-nontrivial states in quantum simulators, especially ultracold atomic gases. Here I give a general answer for Gaussian states and quadratic Lindblad evolution, mostly concentrating on the example of 2D Chern insulator states. I prove a no-go theorem stating that a finite-range Lindbladian cannot induce finite-rate exponential decay towards a unique topological pure state above 1D. I construct a recipe for creating such state by exponentially-local dynamics, or a mixed state arbitrarily close to the desired pure one via finite-range dynamics. I also address the cold-atom realization, classification, and detection of these states. Extensions to other types of topological insulators and superconductors are also discussed.

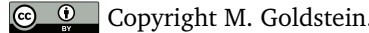

# 1  Introduction

Most of the processes in the world around us occur out of equilibrium. However, the study of many-particle systems has traditionally concentrated on equilibrium or near-equilibrium (linear response) behavior [1]. Part of the reason for this situation is that nonequilibrium effects are typically seen as a nuisance, or even as being detrimental to the appearance of strongly-correlated quantum physics. Indeed, quantum-coherent many-body effects are usually properties of the ground and low-lying states of a system. Nonequilibrium driving tends to decohere a system as well as to excite it away from its ground state, and thus to destroy subtle quantum phenomena [1, 2]. However, it has been realized recently that driven-dissipative dynamics can potentially be used as a *resource*, to drive a system towards nontrivial quantum-correlated states [3–17].

An interesting class of quantum-correlated states are those distinguished by topology [18]. A topologically-nontrivial state of matter cannot be deformed into a trivial one without crossing a quantum phase transition and closing a gap [19]. As has been realized more recently, both theoretically [20–22] and experimentally [23,24], such states may occur even in relatively simple systems, such as band insulators and mean-field superconductors [25–27]. In most cases, symmetries (e.g., time-reversal and/or particle-hole conjugation) are needed to furnish the topological protection. At the boundary between two noninteracting topologically-different systems (e.g., between a topologically-nontrivial system and the vacuum) gapless edge modes must exist, which are immune to localization by disorder and to gapping by any perturbation which is not strong enough to close the bulk gap and does not break the relevant symmetries. For quadratic fermionic models, a complete topological classification of the possible states has been achieved [28–31]. Topological states are also characterized by quantized response functions, such as the Hall conductivity for Chern insulators (integer quantum Hall states on a lattice) [26, 27].

While topological phases of matter were originally discussed in the context of solid-state systems, they may be realized (or "quantum simulated" [32]) in other systems as well. A notable example is provided by ultracold atomic gases [33, 34], where one may introduce an optical lattice or mimic the effects of gauge fields or spin-orbit interaction. Moreover, the ability to probe these systems optically allows to extract parameters which are not measurable in traditional condensed matter systems. Hence, there has been a flurry of experimental and theoretical work aimed at using these unique properties for the implementation and study of various topological models [35–44]. Naturally, there are dissipative contributions to the dynamics of these systems, such as environmental noise, the opposing cooling equipment, and escape/leakage of the particles (atoms, photons, etc.); much effort is directed at minimizing them [32–34]. As mentioned above, this attitude towards nonequilibrium phenomena has

recently changed: It has been theoretically shown that, by proper engineering, one can create systems with *purely-dissipative dynamics* of the Lindblad type [2], which are driven towards quantum correlated steady states [3–5, 11]. Thus, instead of having to tweak the Hamiltonian of the system as well as to couple it to a bath to drive it towards the ground state, one could arrive at the same state by the coupling to the reservoir only, without any internal Hamiltonian dynamics[1].

Can the states achieved in this way be topologically nontrivial? Previous work concentrated on chiral *p*-wave superconductors in 1D and 2D, and provided intriguing results: In the former case it was found that a pure topologically-nontrivial dark state can be realized in a concrete model of ultracold fermions coupled to a bosonic environment, and that this state is accompanied by the appearance of decoupled Majorana modes at the ends of the system, as in equilibrium [7, 15]. In the latter case it was shown that the pure dark state so generated is actually topologically trivial (though decoupled Majorana modes still exist at the centers of vortices) [8, 9]. Modification of the procedure allowed to arrive at topologically-nontrivial state, which is however of low-purity and thus far from the corresponding equilibrium state [13, 16]. The limitations on attaining topological order dissipatively were also studied [6, 10, 14, 17]. In the context of superconducting nanostructures it has been shown how augmenting Hamiltonian dynamics by bath engineering may aid in stabilizing conventional (non-dissipative) fractional quantum Hall states [12]. The scope and generality of these results, as well as the situation for other types of systems, remains unclear. The issue of topological classification of mixed nonequilibrium steady states is also far from resolved [48–56].

In this work I give a general answer to these questions in the context of fermions with quadratic driven-dissipative Lindblad dynamics, using as the primary example Chern-insulator/lattice integer quantum Hall states, that is, 2D systems in symmetry class A [25–27]. I formulate and prove a no-go theorem, which shows that if the Liouvillian super-operator is composed of terms with finite range in real space, exponential decay (at a finite rate in the thermodynamic limit) in time towards a pure steady state with nonzero Chern number (or any other topological state above 1D) cannot be attained. However, a mixed steady state as close as desired to such a pure state is possible. Alternatively, a pure steady state could be reached if the Liouvillian is exponentially (or even faster)-local in real space. I present a concrete recipe for the implementation of the scenarios which are allowed by this theorem.

A possible physical system for the implementation of the recipe is schematically depicted in Fig. 1: Ultracold atoms occupy an optical lattice, which may be modified so as to completely suppress Hamiltonian tunneling dynamics, in manners to be specified below. Rather, the lattice is coupled in a local fashion (e.g., optically) to reservoirs which can supply or take atoms out of the system [2]. I show below how to properly tune these couplings according to the mentioned recipe, so that the induced dissipative dynamics may drive the system to a steady state with the special properties specified above. I also explain the topological classification of the resulting (possibly mixed) state *vis-à-vis* the previous discussion in the literature, and go beyond it by proposing an observable allowing to detect the quantized topological index in a cold-atom experiment.

The paper is organized as follows: After setting the notation and posing the main results as a theorem in Sec. 2, I present the general recipe in Sec. 3, and then show in Sec. 4 that the recipe cannot be improved via the no-go part of the theorem. I then present an example and a concrete cold-atom setup for the realization of the recipe in Sec. 5, and explain how a topological index could be defined and measured in Sec. 6. Finally I conclude in Sec. 7 with

---

[1]Here the aim is to correctly account for the conservation of probability and thus have a steady state, as opposed to using effective non-Hermitian Hamiltonians [45]. This is also different from the Floquet engineering approach [46, 47], which can modify the Hamiltonian but cannot easily control the resulting state.

[2]As shown below, one may forgo the former reservoir, which might be more challenging to realize, at the price of replacing the steady state with a long-lived state.

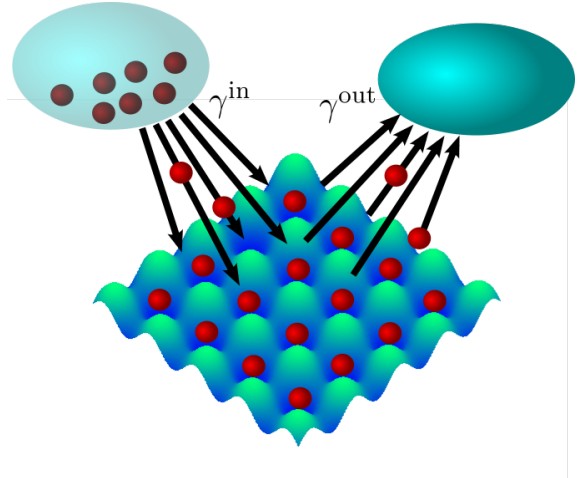

Figure 1: Schematic depiction of a possible realization of the system considered in this work: Fermionic atoms (red spheres) occupy an optical lattice. The lattice is engineered to suppress ordinary tunneling. Instead, the system is connected (e.g., optically) to reservoirs (elliptic blobs) by local couplings (black arrows). The reservoirs exchange atoms with the system and drive it towards a steady state with specific properties, such as being topologically nontrivial.

a discussion of the extension of the results to other topological classes, as well as of future research directions.

## 2  Preliminaries and statement of the main results

Let me start with some preliminaries. The state of a system coupled to an environment is described by a density matrix, $\rho$. When the coupling to the reservoir is weak, the reservoir is much larger than the system (and therefore is negligibly-affected by the system), and has much faster dynamics than the system (conditions which are widely applicable in practice in optically-driven systems, especially ultracold gases), one can integrate out the environment and arrive at a Markovian (memoryless) Lindblad master equation for the density matrix of the system [2]:

$$\partial_t \rho = \hat{\mathcal{L}}[\rho] \equiv -i[H, \rho] + \sum \frac{\gamma_{ij}}{2} \left[ 2L_i \rho L_j^\dagger - L_j^\dagger L_i \rho - \rho L_j^\dagger L_i \right], \tag{1}$$

where the Liouville superoperator $\hat{\mathcal{L}}$ contains two contributions. The first one describes the Hamiltonian part of the evolution, which may include "Lamb shift" corrections due to the coupling to the bath. The second one is the dissipative part. $L_i$ are known as the "Lindblad operators", with a corresponding nonnegative Hermitian rate matrix $\gamma_{ij}$, whose eigenvalues are the various decay and decoherence rates. This is the most general Markovian equation that $\rho$ could obey, which would preserve its Hermiticity, positivity, and overall normalization, $\text{Tr}(\rho) = 1$.

In this work we are going to concentrate on quadratic Lindblad dynamics. For an arbitrary number of baths, if the total number of fermions in the system and the baths is conserved (but fermions may be exchanged between the system to the baths), the most general Lindblad

equation assumes the form

$$\partial_t \rho = \sum_{m,m'} \Big\{ -i[\tilde{h}^S_{mm'} a^\dagger_m a_{m'}, \rho] \tag{2}$$

$$+ \frac{\gamma^{\text{out}}_{mm'}}{2} \big[ 2a_m \rho_S a^\dagger_{m'} - a^\dagger_{m'} a_m \rho_S - \rho_S a^\dagger_{m'} a_m \big] + \frac{\gamma^{\text{in}}_{mm'}}{2} \big[ 2a^\dagger_m \rho_S a_{m'} - a_{m'} a^\dagger_m \rho_S - \rho_S a_{m'} a^\dagger_m \big] \Big\},$$

where $a_m$ is an annihilation operator of a fermion in a single particle state $m$, $\tilde{h}^S_{mm'}$ the Hermitian single-particle system Hamiltonian (which may include corrections due to the coupling to the baths), and $\gamma^{\text{out}}_{mm'}$ and $\gamma^{\text{in}}_{mm'}$ are nonnegative Hermitian rates of particles leaving and entering the system, respectively. For completeness the derivation is presented in Appendix A. Although I will not dwell much on this case, let me mention that when pairing is present, one should add to the Hamiltonian terms of the form $\Delta_{mm'} a_m a_{m'} +$h.c., and similarly add Lindblad terms of the form $\gamma^{\text{pair}}_{mm'}[2a_m \rho_S a_{m'} - a_{m'} a_m \rho_S - \rho_S a_{m'} a_m]/2+$h.c., with antisymmetric matrices $\Delta_{mm'}$ and $\gamma^{\text{pair}}_{mm'}$.

Let me now pose the main question of this study: Can one engineer quadratic Lindblad dynamics for a many-body fermionic system with the above general form which obeys the following requirements:

- Unique steady state $\rho_{ss}$;

- Finite gap in the spectrum of the Liouville superoperator $\hat{\mathcal{L}}$, ensuring that the steady state is approached (in terms of few-particle observables) exponentially fast in time, at a finite rate which independent of the system size [3]. The finite gap also serves to suppress the effects of small perturbations, as in equilibrium;

- Finite range, namely, in a real space representation the matrices $\tilde{h}$, $\gamma^{\text{out}}$ $\gamma^{\text{in}}$, as well as $\Delta$ and $\gamma^{\text{pair}}$, have elements that only connect real space orbitals whose distance is below some upper bound. This is to be contrasted with "local", where elements connecting orbitals at any distance are allowed, provided they decay at least exponentially fast with the distance;

where the resulting steady state $\rho_{ss}$ is:

- Pure, that is, can be written as $\rho_{ss} = |\Psi_d\rangle\langle\Psi_d|$ in terms of a pure dark state $|\Psi_d\rangle$;

- Topologically nontrivial, that is, the dark state $|\Psi_d\rangle$ is characterized by a nonzero topological index, e.g., the Chern number in 2D.

As I will show, one has to give up the strict realization of at least one of these desiderata. This can be stated more formally as follows:

**Theorem.** *A finite range quadratic 2D fermionic Lindbladian cannot have a pure unique steady state which is approached exponentially fast in time at a finite rate independent of the system size, if the pure steady state has nonzero Chern number. However, such a steady state is possible for a local Lindbladian. Alternatively, with a finite range Lindbladian one may obtain a mixed steady state which is approached exponentially fast in time at a finite rate, and which is arbitrarily close (exponentially fast in the range) to a pure steady state that has nonzero Chern number.*

---

[3] This implies a finite mixing rate, for which a finite gap is generally only a necessary, but not sufficient condition. But this issue does not arise for the type of dynamics considered in this work [57]. Let me also note that the time required for arbitrary observables to reach their steady state value, or, equivalently, for the $\ell_1$ distance of the density matrix from $\rho_{ss}$ to fall below some fixed error, would scale logarithmically with the system size, as discussed in Sec. 3 below.

I will start by giving in Sec. 3 a recipe for the realization of either of the allowed cases, which is physically motivated by the possibilities offered in cold-atom systems, of the type depicted schematically in Fig. 1. Then I prove in Sec. 4 the impossibility ("no-go") of the excluded case, showing that the recipe is the best one can indeed achieve. For concreteness, I will concentrate on 2D systems in symmetry class A [25–27], that is, lattice quantum Hall/Chern insulators, but will comment on other symmetry classes along the way and in the Conclusions, Sec. 7.

## 3 Recipe

I will now describe a recipe for realizing a topologically-nontrivial state of noninteracting fermions as a unique steady state of local dynamics obeying the constraints allowed by the Theorem just stated. As we will see, purely-dissipative dynamics [no Hamiltonian part, $H = 0$ in Eq. (1)], will be sufficient. I will assume that the topologically-nontrivial state in question may arise as the ground state (with the lowest band filled) of some finite-range (to be relaxed below to just local) tight-binding *reference Hamiltonian* of fermions on a clean $d$-dimensional lattice (concrete examples will be given below)[4]:

$$H^{\text{ref}} = \sum_{\mathbf{i},\mathbf{i}',\mu,\mu'} h^{\text{ref}}_{\mu\mu'}(\mathbf{i}-\mathbf{i}') a^{\dagger}_{\mathbf{i}\mu} a_{\mathbf{i}'\mu'}, \tag{3}$$

where $\mathbf{i}, \mathbf{i}'$ label lattice unit cells, and $\mu, \mu' = 1, \ldots, n_s$ is a basis of states within a unit cell (possibly including spin), and where $a_{\mathbf{i}\mu}$ obey standard fermionic anticommutation relations[5]. One may then Fourier transform to operators $a_{\mathbf{k}\mu}$ with a given crystal momentum $\mathbf{k}$, and diagonalize the resulting $\mathbf{k}$-dependent $n_s \times n_s$ Hamiltonian matrix $h^{\text{ref}}_{\mu\mu'}(\mathbf{k})$ to find the eigenstates and eigenenergies ($\lambda = 1, \ldots, n_s$ is the band index),

$$H^{\text{ref}} = \sum_{\mathbf{k},\lambda} \varepsilon^{\text{ref}}_{\lambda}(\mathbf{k}) a^{\dagger}_{\mathbf{k}\lambda} a_{\mathbf{k}\lambda}. \tag{4}$$

Let us first assume that the lowest band of the reference Hamiltonian is flat [dispersionless, cf. Fig. 2(a)] in which case we can set $\varepsilon_1(\mathbf{k}) = 0$, without loss of generality; later on we will relax this assumption. Suppose now that the lattice in question has been constructed, e.g., optically in an ultracold atomic system, but that the Hamiltonian tunneling dynamics has been suppressed (by means to be described in Sec. 5 below), cf. Fig. 1. A many-body state describing fermions filling up the lowest band ($\lambda = 1$) completely and leaving the other bands ($\lambda \geq 2$) empty can then be reached from a trivial initial state via finite-range purely-dissipative dynamics. The key idea is to implement not the reference Hamiltonian itself, but rather purely-dissipative dynamics induced by modification of the reference Hamiltonian. This modification involves introducing two internal states of the fermions (e.g., hyperfine states in cold atoms), one which is fully trapped by a confining potential and one which is not trapped in at least one direction, with annihilation operators $a_{\mathbf{i}\mu}$ and $b_{\mathbf{i}\mu}$, respectively. The modified Hamiltonian is then similar to the reference Hamiltonian, but with each tunneling term being replaced by an *optically-assisted hopping term* which also *modifies the internal state of the atom* from trapped to untrapped or vice versa. In the rotating frame this amounts to:

$$H^{\text{out}} = \sum_{\mathbf{i},\mathbf{i}',\mu,\mu'} h^{\text{ref}}_{\mu\mu'}(\mathbf{i}-\mathbf{i}') b^{\dagger}_{\mathbf{i}\mu} a_{\mathbf{i}'\mu'} + \text{h.c.} = \sum_{\mathbf{k},\lambda} \varepsilon^{\text{ref}}_{\lambda}(\mathbf{k}) b^{\dagger}_{\mathbf{k}\lambda} a_{\mathbf{k}\lambda} + \text{h.c.} \tag{5}$$

---

[4]Let me stress again that the reference Hamiltonian should not be confused with the Hamiltonian in Eqs. (1) or (2), which has just been set to zero.

[5]The assumption of translational invariance is not really necessary, and is mainly used to simplify the presentation.

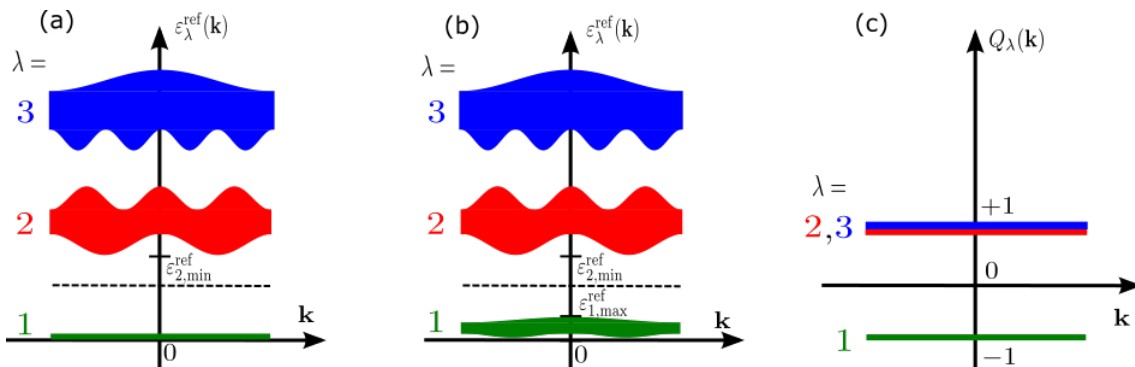

Figure 2: (a) Illustration of a band structure $\varepsilon_\lambda^{\text{ref}}(\mathbf{k})$ with a lowest flat band at zero energy. For clarity, the band structure is projected onto one direction in $\mathbf{k}$ space. Different bands are denoted by $\lambda = 1, 2, 3$ and depicted by different colors. The dashed line denotes the position of the chemical potential. (b) As in (a), but here the lowest band ($\lambda = 1$, green) has finite but small width with respect to the gap to the next band, so that the flatness ratio is small, $\varepsilon_{1,\text{max}}^{\text{ref}}/\varepsilon_{2,\text{min}}^{\text{ref}} \ll 1$. (c) The operator $Q(\mathbf{k})$ used in topological classification [28–31] is the result of flattening the dispersion $\varepsilon_\lambda^{\text{ref}}(\mathbf{k})$ by a continuous deformation, which replaces all the energies above the Fermi energy (here, bands $\lambda = 2, 3$) by $+1$ and all energies below the Fermi energy (the lowest band $\lambda = 1$) by $-1$, without modifying the eigenfunctions.

This Hamiltonian must be supplemented by another term, which includes the confinement of the $a$ fermions, e.g., in the perpendicular direction for a 2D optical lattice, and the possibility of motion of the $b$ atoms along that direction. The latter can be modeled, for example, by assuming that the 2D lattice just introduced is one layer in a 3D lattice. Denoting the crystal momentum in that direction by $q_z$ ($\mathbf{k}$ is still the in-plane momentum) we define the annihilation operators $b_{\mathbf{i}\mu q_z}$, such that $b_{\mathbf{i}\mu} = \sum_{q_z} b_{\mathbf{i}\mu q_z}/\sqrt{N_z}$, $N_z$ being the size of the system (number of sites) in the $z$ direction. Correspondingly we add to the Hamiltonian the term

$$H^{\text{bath}} = \sum_{\mathbf{i},\mu,q_z} [\varepsilon_z(q_z) - \varepsilon_0]\, b_{\mathbf{i}\mu q_z}^\dagger b_{\mathbf{i}\mu q_z}, \tag{6}$$

where $\varepsilon_z(q_z)$ is the dispersion due to the motion in the $z$ direction, and $\varepsilon_0 > 0$ is the detuning between the atomic $a$-$b$ transition frequency (including zero point motion energies) and the frequency of the light inducing the optically-assisted hopping, that is, the residual 1D kinetic energy of the atom after an $a \to b$ transition. I will assume that $b$ atoms that escape the system run away to infinity without ever returning, i.e., that the $b$ fermions effectively constitute a bath with a negative infinite chemical potential. Hence, the bath density matrix $\rho_E$ obeys $\text{Tr}(\rho_E b_{\mathbf{i}\mu q_z}^\dagger b_{\mathbf{i}'\mu'q_z'}) = 0$ and $\text{Tr}(\rho_E b_{\mathbf{i}\mu q_z} b_{\mathbf{i}'\mu'q_z'}^\dagger) = \delta_{\mathbf{i}\mathbf{i}'}\delta_{\mu\mu'}\delta q_z q_z'$. If the escape rate of the $b$ fermions is much faster than the optical transition rates, every optically-assisted hopping event leads to a loss of an atom. The conditions mentioned above for the validity of the Lindblad Markov equation are thus fulfilled. Then, we may integrating out the $b$ fermions [using the second form of Eq. (5)] in the manner described in Appendix A and derive the following

Lindbladian for the trapped $a$ atoms [6]:

$$\hat{\mathcal{L}}^{\text{out}}\rho = \sum_{\mathbf{k},\lambda} \frac{\gamma_\lambda^{\text{out}}(\mathbf{k})}{2} \left(2a_{\mathbf{k}\lambda}\rho a_{\mathbf{k}\lambda}^\dagger - a_{\mathbf{k}\lambda}^\dagger a_{\mathbf{k}\lambda}\rho - \rho a_{\mathbf{k}\lambda}^\dagger a_{\mathbf{k}\lambda}\right) \tag{7}$$

$$= \sum_{\mathbf{i},\mathbf{i}',\mu,\mu'} \frac{\gamma_{\mu\mu'}^{\text{out}}(\mathbf{i}-\mathbf{i}')}{2} \left(2a_{\mathbf{i}'\mu'}\rho a_{\mathbf{i}\mu}^\dagger - a_{\mathbf{i}\mu}^\dagger a_{\mathbf{i}'\mu'}\rho - \rho a_{\mathbf{i}\mu}^\dagger a_{\mathbf{i}'\mu'}\right),$$

where, by the Fermi golden rule, $\gamma_\lambda^{\text{out}}(\mathbf{k}) = 2\pi\nu_0[\varepsilon_\lambda^{\text{ref}}(\mathbf{k})]^2$, and thus $\gamma_{\mu\mu'}^{\text{out}}(\mathbf{i}-\mathbf{i}') = 2\pi\nu_0\sum_{\mathbf{j},\mu''}h_{\mu\mu''}^{\text{ref}}(\mathbf{i}-\mathbf{j})h_{\mu''\mu'}^{\text{ref}}(\mathbf{j}-\mathbf{i}')$. Here $\nu_0$ is the 1D ($z$ direction) local density of states per unit cell of the untrapped atomic internal state $b$ emitted with kinetic energy $\varepsilon_0$; a concrete expression will be given in Sec. 5 below.

By our previous assumptions [$\varepsilon_1(\mathbf{k}) = 0$ and separated by a finite gap from the other $\varepsilon_{\lambda\geq2}(\mathbf{k}) = 0$], the escape rate is finite for all the higher bands ($\lambda \geq 2$) but is exactly zero for the lowest band ($\lambda = 1$). Thus, if one starts at the trivial, completely filled state for the $a$ atoms, it will exponentially decay towards the desired pure many body dark state $|\Psi_d\rangle$, in which the lowest band is completely filled and all the others are empty. As a matter of fact, the nature of the final state does not depend on the Markovian approximation used to derive Eq. (7), and stems already from the Hamiltonian (5). This Hamiltonian coupled all the $a$ atoms to the reservoir (and thus dooms them to leave the system sooner or later), except those that occupy the states of the lowest band of the reference Hamiltonian[7]. Let me note that, for the purely evaporative dynamics presented so far, the final state is not unique and depends on the initial conditions, since if one of the states in the $\lambda = 1$ band is initially empty, it will not be refilled. To remedy this I would later on introduce a "refilling" reservoir, which could similarly be engineered to fill in only states in the $\lambda = 1$ band. But before going into that, let us first continue the discussion of the evaporative dynamics, Eq. (7).

The success of the above evaporative procedure crucially depends on the flatness of the desired $\lambda = 1$ band of the reference Hamiltonian. However, as has been appreciated for some time [58–63] but proven only quite recently [64–66], a topologically-nontrivial (having nonzero Chern number in the absence of symmetries, or arising from a reference Hamiltonian with some symmetry and having a nontrivial topological index appropriate for that symmetry class) flat band cannot arise in 2D and above from a finite-range gapped reference Hamiltonian[8], that is, with matrix elements $h_{\mu\mu'}^{\text{ref}}(\mathbf{i}-\mathbf{i}')$ which vanish for $\|\mathbf{i}-\mathbf{i}'\|$ larger than some fixed value. This implies similar range restriction for the Lindbladian (7) built out of it. In the next subsection I will show that this is not a peculiarity of the current construction, but rather a result of a general no-go theorem. Here I will instead discuss what one may still achieve and how.

One possibility is to forgo finite range and content ourselves with locality, that is, $h_{\mu\mu'}^{\text{ref}}(\mathbf{i}-\mathbf{i}')$, and hence $\gamma_{\mu\mu'}^{\text{out}}(\mathbf{i}-\mathbf{i}')$, which decay rapidly with $\|\mathbf{i}-\mathbf{i}'\|$. An exponential decay is easily seen to be sufficient: One may replace the Hamiltonian by a projector onto the bands $\lambda \geq 2$, which, due to the gap in the original Hamiltonian, will have exponentially-decaying matrix elements in real space [67–69]. In fact, even faster decay is possible, as is exemplified by the Kapit-Mueller Hamiltonian [70], which is characterized by matrix elements decaying exponentially with the *square* of the distance, and leads to a lowest flat band with nonzero Chern number.

---

[6]In addition, there is a Lamb shift contribution generating a Hamiltonian for the $a$ fermions, as described in Appendix A. This Hamiltonian is however also diagonal in the eigenbasis of the reference Hamiltonian. Hence, it does not affect the resulting steady state.

[7]This may be important in practice, since if the escape rate of the $b$ atoms is too large, the escape of the $a$ atoms will be suppressed by a Zeno-like effect.

[8]The gap between the lowest and other bands is necessary for ensuring finite decay rates for all the states not in the lower band, see Eq. (7).

In fact, I am not aware on any bound on the rate of decay, except that $h_{\mu\mu'}^{\text{ref}}(\mathbf{i}-\mathbf{i}')$ cannot have finite support (finite range). This is to be contrasted with the corresponding Wannier functions, which can only decay algebraically with distance for a band with nonzero Chern number [71–73].

Another possibility is to keep the finite range of the terms in the Lindbladian, and instead forgo the requirement of purity of the steady state. A finite-range reference Hamiltonian may have a very narrow topological lowest band, in the sense that its width is much smaller than the gap separating it from the next band, $\lambda = 2$ [58–63]. For convenience we set the zero of energy at the minimum of the lowest band [see Fig. 2(a)–(b)][9]. The narrowness of the lowest band can then be quantified in terms of the flatness ratio, i.e., the ratio of the maximum of the lowest band to the minimum of the second band, $\varepsilon_{1,\max}^{\text{ref}}/\varepsilon_{2,\min}^{\text{ref}} \ll 1$. In this case the Lindbladian (7) strongly depletes the modes belonging to the higher ($\lambda \geq 2$) bands of the reference Hamiltonian [at rates $\geq \gamma_{2,\min}^{\text{out}} \equiv 2\pi\nu_0(\varepsilon_{2,\min}^{\text{ref}})^2$], while only weakly affecting the lowest band [$\lambda = 1$, which is depleted at rates $\leq \gamma_{1,\max}^{\text{out}} \equiv 2\pi\nu_0(\varepsilon_{1,\max}^{\text{ref}})^2$]. The depletion rate ratio thus scales as $(\varepsilon_{2,\min}^{\text{ref}}/\varepsilon_{1,\max}^{\text{ref}})^2 \gg 1$, the *square* of the inverse flatness ratio of the reference Hamiltonian.

I now add a competing term,

$$
\begin{aligned}
\hat{\mathcal{L}}^{\text{in}}\rho &= \frac{\gamma^{\text{in}}}{2}\sum_{\mathbf{i},\mu}\Big(2a_{\mathbf{i}\mu}^{\dagger}\rho a_{\mathbf{i}\mu} - a_{\mathbf{i}\mu}a_{\mathbf{i}\mu}^{\dagger}\rho - \rho a_{\mathbf{i}\mu}a_{\mathbf{i}\mu}^{\dagger}\Big) \\
&= \frac{\gamma^{\text{in}}}{2}\sum_{\mathbf{k},\lambda}\Big(2a_{\mathbf{k}\lambda}^{\dagger}\rho a_{\mathbf{k}\lambda} - a_{\mathbf{k}\lambda}a_{\mathbf{k}\lambda}^{\dagger}\rho - \rho a_{\mathbf{k}\lambda}a_{\mathbf{k}\lambda}^{\dagger}\Big),
\end{aligned}
\tag{8}
$$

which tries to populate all the lattice states at a constant rate $\gamma^{\text{in}}$. This could be realized by using yet another atomic species/internal state $c$ in a setup similar to $b$ except that the $c$ bath is filled with a large number of atoms (effectively making it an infinite chemical potential reservoir), and that the matrix elements coupling it to $a$ in the analogue of Eq. (5) are proportional to the unit matrix, $\delta_{\mathbf{i}\mathbf{i}'}\delta_{\mu\mu'}$. The combined Lindbladian $\hat{\mathcal{L}} = \hat{\mathcal{L}}^{\text{out}} + \hat{\mathcal{L}}^{\text{in}}$ is quadratic, and thus has a Gaussian steady state (a unique one, since $\hat{\mathcal{L}}^{\text{in}}$ has no zero modes [57]), which is fully characterized by the positive and Hermitian single-particle reduced density matrix, $G_{\mu\mu'}^{ss}(\mathbf{i}-\mathbf{i}') \equiv \text{Tr}(\rho_{ss}a_{\mathbf{i}\mu}^{\dagger}a_{\mathbf{i}'\mu'})$. The latter is diagonal in the basis of the eigenmodes of the reference Hamiltonian,

$$
G_{\lambda\lambda'}^{ss}(\mathbf{k},\mathbf{k}') = \text{Tr}\big(\rho a_{\mathbf{k}\lambda}^{\dagger}a_{\mathbf{k}'\lambda'}\big) = \delta_{\lambda\lambda'}\delta_{\mathbf{k}\mathbf{k}'}n_{\lambda}^{ss}(\mathbf{k}),
\tag{9}
$$

with the steady-state mode occupancies $n_{\lambda}^{ss}(\mathbf{k}) = \gamma^{\text{in}}/[\gamma^{\text{in}} + \gamma_{\lambda}^{\text{out}}(\mathbf{k})]$. Choosing $\gamma^{\text{in}}$ in the range $\gamma_{1,\max}^{\text{out}} \ll \gamma^{\text{in}} \ll \gamma_{2,\min}^{\text{out}}$, single-particle states belonging to the lowest band will be almost filled, while the other bands would remain almost empty, as desired. The optimal value of $\gamma^{\text{in}}$ will be of order $(\gamma_{1,\max}^{\text{out}}\gamma_{2,\min}^{\text{out}})^{1/2}$, for which all the deviations of the occupancies of the mixed steady state from those of the pure ground state of the reference Hamiltonian (with chemical potential in the gap between the bands $\lambda = 1$ and $\lambda = 2$) are of the order of the flatness ratio $\varepsilon_{1,\max}^{\text{ref}}/\varepsilon_{2,\min}^{\text{ref}} \ll 1$. The latter decreases exponentially with the range of the terms in the reference Hamiltonian [63,67–69]. This immediately translates into an exponential decrease (with the range of the Lindbladian) of the deviation between few particle observables (observables depending on modes whose number does not scale with the system size) of the mixed steady state and the corresponding pure state values. Therefore, the required range to obtain a fixed deviation scales logarithmically with the desired deviation size, independently of the system

---

[9]Note, though, that it is actually better to shift it downwards, so as to minimize the maximal absolute value of the energy in the lowest band, which determines the evaporation rate, cf. Eq. (7).

size. As for more general observables, they follow the $\ell_1$ distance of the mixed steady state from the pure desired one, which also decays exponentially with the range, but with a prefactor proportional to the system size. Hence, the required range to achieve a fixed deviation of these quantities varies logarithmically with both the desired deviation and the system size.

Furthermore, any deviation of few-particle observables from their steady state value will decay exponentially in time at a finite rate which is independent of the system size, $\gamma_\lambda^{\mathrm{out}}(\mathbf{k}) + \gamma^{\mathrm{in}} \geq \gamma^{\mathrm{in}}$. For example, the mode occupancies would decay as

$$n_\lambda(\mathbf{k}, t) = n_\lambda^{ss}(\mathbf{k}) + \left[ n_\lambda(\mathbf{k}, t) - n_\lambda^{ss}(\mathbf{k}) \right] e^{-[\gamma_\lambda^{\mathrm{out}}(\mathbf{k}) + \gamma^{\mathrm{in}}]t}, \tag{10}$$

which is a particular case of Eq. (12) below. Hence, the time required to achieve a certain fixed deviation varies logarithmically with the desired deviation size, independently of the system size. Again, general observables will follow the $\ell_1$ distance of the density matrix from the steady state $\rho_{ss}$, which will feature a prefactor proportional to the system size. As a result, the time needed for the $\ell_1$ distance to become smaller than some fixed deviation will scale logarithmically with both the desired deviation and the system size. Let me also mention that the finite rate $\geq \gamma^{\mathrm{in}}$ amounts to a finite gap in the gap of the Liouvillian, which would act to protect the system from perturbations (in analogy to the excitation gap in the equilibrium case [25–27]).

It should also be noted that one may create a similar state without invoking the incoming Lindbladian (8). Instead one may start from a completely filled initial state and apply the outgoing Lindbladian (7) for a finite time $\sim (\gamma_{1,\mathrm{max}}^{\mathrm{out}} \gamma_{2,\mathrm{min}}^{\mathrm{out}})^{-1/2}$. The resulting simplification of the setup should be weighted, however, against the non-uniqueness of the steady state, and thus the lack of stabilization of the state after its creation.

Finally, let me come back to the implementation a local (not short-ranged) Lindbladian with a pure unique steady state. For this purpose one would need to improve the construction of the refilling Lindbladian, Eq. (7), and choose a space-dependent $\gamma_{\mu\mu'}^{\mathrm{in}}(\mathbf{i} - \mathbf{i}')$ built out of a flat-band reference Hamiltonian which is a projector into the $\lambda = 1$ band [similar but opposite to the construction of $\gamma_{\mu\mu'}^{\mathrm{out}}(\mathbf{i} - \mathbf{i}')$], which will refill only states belonging to that band. With this one would have $\gamma_{\lambda=1}^{\mathrm{out}}(\mathbf{k}) = \gamma_{\lambda\neq1}^{\mathrm{in}}(\mathbf{k}) = 0$, and hence $n_\lambda^{ss}(\mathbf{k}) = \delta_{\lambda,1}$, corresponding to the desired pure steady state.

## 4 No-go theorem

I will now proceed to prove the negative side of the theorem stated in Sec. 2, namely that one cannot do better than the above recipe. This implies that one cannot achieve exponential decay in time at a finite (system size independent) rate towards a pure dark state with nonzero Chern number in 2D if the Lindbladian has a finite range. I will do so by showing that this would amount to having a finite-range Hermitian operator in 2D featuring flat eigenbands with nonzero Chern number separated by gaps from other bands, which is known to be impossible [64–66]. Similar results hold for other topological classes [66].

To set the stage for the proof, let us examine general quadratic Lindblad dynamics, described by Eq. (2). This gives rise to the following evolution equation for the single-particle density matrix $G$ (defined as in the previous Section but now not necessarily in the steady state):

$$\partial_t G = -\left( \frac{\gamma^{\mathrm{out}} + \gamma^{\mathrm{in}}}{2} - i\tilde{h}^S \right) G - G \left( \frac{\gamma^{\mathrm{out}} + \gamma^{\mathrm{in}}}{2} + i\tilde{h}^S \right) + \gamma^{\mathrm{in}}. \tag{11}$$

The steady state single-particle density matrix is the solution of $\partial_t G^{ss} = 0$. The decay towards

it is given by

$$G(t) = G^{ss} + e^{-\left(\frac{\gamma^{out}+\gamma^{in}}{2} - i\tilde{h}^S\right)t}\left[G(t=0) - G^{ss}\right]e^{-\left(\frac{\gamma^{out}+\gamma^{in}}{2} + i\tilde{h}^S\right)t}. \tag{12}$$

The positivity of $\gamma^{out}$ and $\gamma^{in}$ implies that the real parts of the eigenvalues of $(\gamma^{out}+\gamma^{in})/2 \pm i\tilde{h}^S$ are nonnegative. To ensure a finite decay rate towards the steady state in the thermodynamic limit, the lower bound of these real parts must be positive and finite as the system size goes to infinity, i.e., there should be a finite gap between the real part of the spectrum of $(\gamma^{out}+\gamma^{in})/2 \pm i\tilde{h}^S$ and zero.

With this we can return to the no-go theorem. Its proof consists of two steps:

1. The main one is to show that a pure steady state implies that the positive Hermitian rate matrices $\gamma^{out}$ and $\gamma^{in}$ have flat bands with eigenvalue zero, corresponding to the filled and empty single-particle states, respectively. Since these bands should be topologically nontrivial, if $\gamma^{out}$ and $\gamma^{in}$ have finite range, they are gapless.

2. The second step is to demonstrate that the gaplessness of $\gamma^{out}$ and $\gamma^{in}$ implies that the real part of the spectrum of $(\gamma^{out}+\gamma^{in})/2 \pm i\tilde{h}^S$ is not separated by a finite gap from zero, hence the decay rate towards the steady state vanishes in the thermodynamic limit, in contradiction to our assumptions.

The first step results from considering the steady state. Let us work in the single-particle basis where $G^{ss}$ is diagonal, which is a basis of Bloch states if the system is translationally invariant. Since $\rho_{ss}$ is Gaussian, if it is also pure, the eigenvalues of $G^{ss}$, namely the eigenoccupancies $n_\lambda^{ss}(\mathbf{k})$, are all either 0 or 1, corresponding to empty and filled states, respectively. We will write all the matrices appearing in Eq. (11) in terms of blocks in the empty-filled basis, e.g., $\tilde{h}^S = \begin{pmatrix} \tilde{h}_{00}^S & \tilde{h}_{01}^S \\ \tilde{h}_{10}^S & \tilde{h}_{11}^S \end{pmatrix}$. In this basis $G^{ss} = \begin{pmatrix} 0 & 0 \\ 0 & I \end{pmatrix}$, with $I$ being a unit matrix. Substituting into Eq. (11) we find that $\gamma_{11}^{out} = 0$ and $\gamma_{00}^{in} = 0$, which, due to the positivity of the rate matrices, also implies that $\gamma_{01}^{out} = \gamma_{01}^{in} = 0$ and $\gamma_{10}^{out} = \gamma_{10}^{in} = 0$. In addition we get $\tilde{h}_{01}^S = 0$ and $\tilde{h}_{10}^S = 0$, which will become important in the next step. Thus $\tilde{h}^S = \begin{pmatrix} \tilde{h}_{00}^S & 0 \\ 0 & \tilde{h}_{11}^S \end{pmatrix}$, $\gamma^{out} = \begin{pmatrix} \gamma_{00}^{out} & 0 \\ 0 & 0 \end{pmatrix}$, and $\gamma^{in} = \begin{pmatrix} 0 & 0 \\ 0 & \gamma_{11}^{in} \end{pmatrix}$. This is a natural result: Completely filled (empty) modes are modes which do not couple at all to the outgoing (incoming) bath, and thus form a flat band with eigenvalue zero of the Hermitian matrix $\gamma^{out}$ ($\gamma^{in}$), which we assume to have a finite range. However, since the respective modes have a nonzero Chern number [64–66], this is impossible unless $\gamma^{out}$ ($\gamma^{in}$) is not gapped, i.e., the positive Hermitian block $\gamma_{00}^{out}$ ($\gamma_{11}^{in}$) has eigenvalues which are either zero or vanish in the thermodynamic limit.

I now turn to the second step. By Eq. (12), the last conclusion (gaplessness of $\gamma^{out}$ and $\gamma^{in}$ with respect to zero) implies for purely dissipative dynamics ($\tilde{h}^S = 0$), or when $\tilde{h}^S \neq 0$ but there is only one empty or one filled band, the absence of a finite (system-size-independent) exponential decay rate towards the steady state. The general case of $\tilde{h}^S \neq 0$ and arbitrary number of bands requires a slightly more involved argument. We have observed above that the filled (empty) bands are flat bands with zero eigenvalue of the Hermitian matrix $\gamma^{out}$ ($\gamma^{in}$), whose elements are short-range in real space, and hence are trigonometric polynomials in the $\mathbf{k}$ basis. The filled (empty) bands wavefunctions thus obey linear equations with coefficients which are trigonometric polynomials in $\mathbf{k}$, i.e., these bands form a polynomial bundle in the terminology of Refs. [65, 66]. On the other hand, as mentioned above, a finite decay rate of $G$ towards its steady state value $G^{ss}$ in the thermodynamic limit implies a finite gap between zero and the real part of the spectrum of $(\gamma^{out}+\gamma^{in})/2 \pm i\tilde{h}^S = \begin{pmatrix} \gamma_{00}^{out}/2 \pm i\tilde{h}_{00}^S & 0 \\ 0 & \gamma_{11}^{in}/2 \pm i\tilde{h}_{11}^S \end{pmatrix}$,

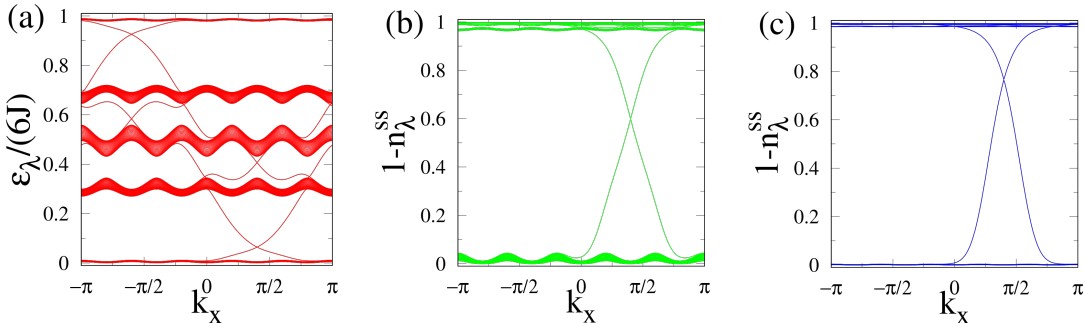

Figure 3: (a) The eigenenergies of the Hofstadter reference Hamiltonian (13) with periodic boundary conditions in the $x$ direction (that is, on a cylinder whose axis is in the $y$ direction) for $\alpha = 1/5$ and $J_0 = 2.93J$ (set to bring the minimal energy to zero). In between the $q = 5$ bands there are chiral midgap edge modes localized at the two ends of the system. (b) The spectrum of the steady state single-particle reduced density matrix, Eq. (9) [$1-n_\lambda^{ss}$ is plotted so that the order of states is as in (a)] resulting from the Lindblad dynamics, Eq. (7)–(8), with $\gamma^{in} = 0.2\pi \nu_0 J^2$. The four upper bands appear together at the top, but are actually separated by small gaps. (c) The same with the reference Hamiltonian (13) containing the next-nearest-neighbor hopping terms of the Kapit-Mueller Hamiltonian [70].

which holds separately for each of its two nonzero blocks. There will then be a finite gap between the spectra of the two blocks of, e.g., $\gamma^{out}/2 + i\tilde{h}^S = \begin{pmatrix} \gamma_{00}^{out}/2 + i\tilde{h}^S 00 & 0 \\ 0 & i\tilde{h}_{11}^S \end{pmatrix}$, since the lower block has purely imaginary eigenvalues. By the arguments of Refs. [65, 66] this would imply that the filled and empty bands form analytic bundles [10]. But since these bundles were argued to be polynomial, they cannot be topologically nontrivial [65,66], in contradiction to our assumptions. This completes the proof. Similar results would hold for other topological classes above 1D, if one enforces that $\gamma^{out}$ or $\gamma^{in}$ obey the respective symmetry [66].

Let us note that one can further extend the model to include cases of "superconducting" reservoirs, where in Eq. (1) the Hamiltonian may contain pairing terms with either two creation operators or two annihilation operators, and in the Lindbladian there could be "pairing" terms where $L_i$ and $L_j^\dagger$ are both creation or both annihilation operators. In that case it would be useful to define Majorana fermions, $f_{i,\mu}^+ = a_{i,\mu} + a_{i,\mu}^\dagger$ and $f_{i,\mu}^- = i(a_{i,\mu} - a_{i,\mu}^\dagger)$, and to use $G_{\mu\mu'}^{\pm\pm}(\mathbf{i}-\mathbf{i}') = \text{Tr}(\rho f_{i,\mu}^\pm f_{i',\mu'}^\pm)$. One may then write down an analogous equation to Eq. (11) and use similar arguments to reach the same conclusion, namely that a topologically-nontrivial pure state with superconducting correlations, such as a chiral $p$-wave (class D [25–27]) state in 2D, cannot be obtained at a finite rate in time as the unique steady state of a gapped finite-range quadratic Lindbladian, but could be created if only locality is imposed. This in retrospect explains the difference found in previous studies between 1D and 2D dissipative pairing systems: In 1D the Kitaev chain [74] has a well-known flat-band limit, allowing a finite-range Lindbladian with a pure topological steady state, as found in Refs. [7–9]. In 2D this is not possible [64–66], so only a mixed steady state can be reached, as surmised in Ref. [13]. The results presented here bear some relation to the tradeoff between purity and short range in tensor network representations of topological states [69].

---

[10]This holds even though $\gamma^{out}/2 + i\tilde{h}^S$ is not hermitian.

# 5 Example: Dissipative Hofstadter model with cold atoms

As an example for the usage of the above technique, I will apply it to realize a lattice integer quantum Hall state. Here the reference Hamiltonian is the Hofstadter model [75], a tight binding model with nearest-neighbor hopping on a square lattice pierced by a magnetic flux,

$$H^{\mathrm{ref}} = \sum_{n_x,n_y} J_0 a^{\dagger}_{n_x,n_y} a_{n_x,n_y} + J e^{i2\pi\alpha n_y} a^{\dagger}_{n_x+1,n_y} a_{n_x,n_y} + J a^{\dagger}_{n_x,n_y+1} a_{n_x,n_y} + \mathrm{h.c.}, \qquad (13)$$

where $\alpha$, the magnetic flux per plaquette in units of the flux quantum, is taken as rational, $\alpha = p/q$ ($p,q$ are integers with no common factor). Fig. 3(a) shows its spectrum on a cylinder with periodic boundary conditions in the $x$ direction for $\alpha = 1/5$. It features $n_s = q = 5$ bands as well as chiral midgap modes localized at the two edges of the system. The flatness ratio for the lowest band is below $4.2 \cdot 10^{-2} \ll 1$.

For the recipe one needs to implement the two-flavor analogue of this Hamiltonian, cf. Eq. (5). There are numerous proposals for engineering the necessary gauge fields artificially in cold-atom systems [35–44], many of which could be adapted to create the purely dissipative dynamics described here. As an example I will concentrate on the construction suggested in [76, 77]. In the current context it would involve fermionic two-level atoms on a 3D tetragonal optical lattice, with lattice constants $d$ and $d_z$ in the xy plane and z direction, respectively. The $b$ atoms are free to propagate in 3D, but the $a$ atoms are confined to a single 2D plane with a steep enough confinement potential that the confinement level spacing is much larger than all the other energy scales introduced below (except $\omega_0$). The lattice is modulated in the $x$ and $y$ directions in the way depicted in Fig. 4. Employing a tight-binding single-band description, this corresponds to a Hamiltonian

$$H_V = \sum_{n_x,n_y} V_{n_x,n_y} a^{\dagger}_{n_x,n_y} a_{n_x,n_y} + \sum_{n_x,n_y,q_z} \left[ \varepsilon_z(q_z) + V_{n_x,n_y} + \omega_0 \right] b^{\dagger}_{n_x,n_y,q_z} b_{n_x,n_y,q_z}, \qquad (14)$$

where $a_{n_x,n_y}$ annihilates an $a$ atom in the Wannier orbital $w_a(x - n_x d, y - n_y d, z)$, including the confinement in the $z$ direction, while $b_{n_x,n_y,q_z}$ annihilates a $b$ atom in the state $\sum_{n_z} w_b(x - n_x d, y - n_y d, z - n_z d_z) e^{i n_z q_z d_z}/\sqrt{N_z}$, localized in the 2D plane but propagating with crystal momentum $q$ along the $z$ direction, which is $N_z$-layers long. $\omega_0$ is the $a \to b$ transition frequency (including differences in zero point motion energies), $\varepsilon_z(q) = 4J_z \sin^2(qd/2)$, and the modulated potential is $V_{n_x,n_y} = [\delta_{n_x \equiv 1 \,(\mathrm{mod}\, 2)} + \delta_{n_y \equiv 1 \,(\mathrm{mod}\, 2)}]\varepsilon_y + [\delta_{n_x \equiv 2 \,(\mathrm{mod}\, 4)} - \delta_{n_x \equiv 1 \,(\mathrm{mod}\, 4)}]\varepsilon_x$. The resulting mismatch in energies between neighboring sites prevents usual hopping for both the $a$ and $b$ states, provided the bare hoppings in the 2D plane as well as $J_z$ are all much smaller than both $\varepsilon_x$ and $\varepsilon_y$.

One may now turn on lasers with frequencies chosen so as to allow photon-assisted nearest-neighbor tunneling events which change the internal states of the atoms ($a$ and $b$, or trapped and un-trapped, depicted as blue and orange, respectively, in Fig. 4). A laser beam with 3D wavevector $\mathbf{k}_L = (k_{Lx}, k_{Ly}, k_{Lz})$, frequency $\omega_L$, and Rabi frequency (amplitude) $\Omega_L$ allows for transitions between $a_{n_x n_y}$ and $b_{n'_x n'_y} \equiv \sum_{q_z} b_{n'_x n'_y q_z}/\sqrt{N_z}$ provided that the frequency mismatch obeys $0 < \omega_L - (\omega_0 + V_{n'_x n'_y} - V_{n_x n_y}) < 4J_z$. One may pick the following frequencies, as indicated in Fig. 4:

- $\varepsilon_0 + \omega_0 \pm \varepsilon_y$ (dotted arrows), allowing nearest-neighbor state flip hopping along $y$, as well as along $x$ between $n_x \equiv 3 \,(\mathrm{mod}\, 4)$ and $n_x + 1$;

- $\varepsilon_0 + \omega_0 \pm (\varepsilon_y - \varepsilon_x)$ (solid), allowing nearest-neighbor state flip hopping along $x$ between $n_x$ even and $n_x + 1$;

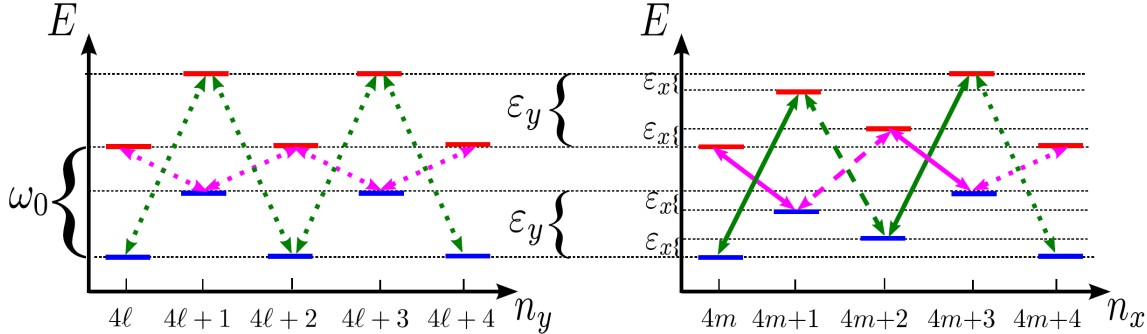

Figure 4: Lattice modulation scheme along the $y$ (left) and $x$ (right) directions corresponding to Eq. (14). Each lattice site may contain fermionic atoms with two internal states [blue and red, the latter not being trapped, corresponding to $a$ and $b$ in Eq. (16)] whose energies differ by $\omega_0$. A period-2 modulation with amplitude $\varepsilon_y$ is applied in *both* directions. An additional period-4 modulation with amplitude $\varepsilon_x$ is applied in the $x$ direction only. Optically assisted hopping is created by 6 laser beams with appropriate frequencies (green/pink solid/dashed/dotted arrows), whose directions are chosen so as to reproduce the phase factors in the Hofstadter Hamiltonian, Eq. (13) [76, 77]; an additional beam with frequency $\omega_0$ (not shown) allows for an on-site term. This gives rise to the Hofstadter system-bath coupling Hamiltonian (16), and thus to the evaporative Lindbladian (7).

- $\varepsilon_0 + \omega_0 \pm (\varepsilon_y - 2\varepsilon_x)$ (dashed), allowing nearest-neighbor state flip hopping along $x$ between $n_x \equiv 1 \,(\mathrm{mod}\, 4)$ and $n_x + 1$;

- A laser of frequency $\omega_0 + \varepsilon_0$ (not depicted in Fig. 4) is used to induce onsite transitions.

Here $0 < \varepsilon_0 \ll \varepsilon_x, \varepsilon_y$ is a small detuning.

For each laser, the resulting optically-induced transition amplitude is

$$J_L = \frac{\Omega_L}{2} e^{ik_{Lx}n_x d + ik_{Ly}n_y d} \int \mathrm{d}x \int \mathrm{d}y \int \mathrm{d}z \, w_a^*(x - n_x d, y - n_y d, z)$$
$$\times w_b(x - n'_x d, y - n'_y d, z) e^{ik_{Lx}x + ik_{Ly}y + ik_{Lz}z}. \tag{15}$$

One may therefore use the Rabi frequencies to control the magnitudes of the various $J_L$ and make them equal to $J$ for all the beams except the onsite one, for which it should equal $J_0$. In addition, the directions of the lasers should be set so that their wavevectors are in the $yz$ plane with the appropriate projection onto the $y$ axis to give the tunneling matrix elements along the $x$ axis the phase specified in the reference Hamiltonian (13), exactly as in [76,77][11]. In the rotating frame with respect to the $V_{n_x n_y}$ and $\omega_0$ terms in Eq. (14) one thus obtains the following Hamiltonian:

$$H^{\mathrm{ref}} = \sum_{n_x, n_y, q_z} J_0 a^\dagger_{n_x, n_y} b_{n_x, n_y, q_z} + J e^{i2\pi\alpha n_y} a^\dagger_{n_x+1, n_y} b_{n_x, n_y, q_z} + J a^\dagger_{n_x, n_y+1} b_{n_x, n_y, q_z} + \mathrm{h.c.}$$
$$+ [\varepsilon_z(q_z) - \varepsilon_0] b^\dagger_{n_x, n_y, q_z} b_{n_x, n_y, q_z}. \tag{16}$$

This is exactly the sum of Eqs. (5) and (6) for the specific reference Hamiltonian (13). Integrating out the $b$ fermions thus gives the desired outgoing Lindbladian, Eq. (7) with $\nu_0 = 1/[\pi\sqrt{\varepsilon_0(4J_z - \varepsilon_0)}]$, the local density of states per unit cell due to the $z$ direction motion.

---

[11]With the scheme suggested here there will also be nonzero phases for tunnelings in the $y$-direction. However, these depend only on $y$, hence can be eliminated by a gauge transformation.

As advocated in Ref. [77], this construction is best-suited to two-electron atoms, such as $^{171}$Yb. One may use the clock transition states, that is, the ground state $^1S_0$, and the long-lived first excited state $^3P_0$, as the $a$ and $b$ states, respectively. Most of the parameters could be set as suggested there, with a couple of important differences: (i) In the system (xy) plane both the $a$ and $b$ states should occupy the same square lattice, hence one may use the magic wavelength to produce it (as assumed for simplicity in Fig. 4). However, one may employ any wavelength for which the polarizability of the two states has the same sign, which would allow to go further away from atomic resonances and reduce heating; (b) For the confinement in the $z$ direction one should use the wavelength for which the $b$-state polarizability vanishes (which was the assumption in the previous discussion), or any wavelength for which the $b$-state polarizability is opposite to that of the $a$ state, thus confining the latter but not the former atoms. The lattice modulation in the $z$ direction could be created by any wavelength with non-vanishing $b$ polarizability.

One may then set the 3D square optical lattice depth to be of the order of $10E_R$, where $E_R \sim 2\pi \cdot 1$ kHz is the free space recoil energy of an atom due to the absorption or emission of a single photon at the relevant wavelength. The corresponding ordinary tunneling amplitudes would then be of order $\sim 5 \cdot 10^{-2}E_R$. Choosing the modulations $\varepsilon_x, \varepsilon_y$ of order of a few $E_R$ would then strongly suppress ordinary tunneling in the xy plane. On the other hand, choosing the laser amplitudes about 3 times smaller than the lattice band spacing (to prevent exciting higher bands) could give optically assisted tunneling amplitudes of order $J \sim 10^{-2}E_R$. As follows from Eq. (7) and the subsequent discussion, by reducing $\varepsilon_0$, that is, choosing frequencies such that the residual kinetic energy of the $b$ atoms is small, one may increase $\nu_0 = 1/[\pi\sqrt{\varepsilon_0(4J_z - \varepsilon_0)}]$ and thus the rates $\gamma^{\text{out}}$. The only limitation is that $\gamma^{\text{out}}$ should be small enough with respect to $E_R$ so as not to mix in unwanted states such as higher bands of the optical lattice. Thus one may achieve $\gamma^{\text{out}} \sim 2\pi\nu_0 J^2 \sim 10^{-1}E_R$.[12] The corresponding timescale would be a couple orders of magnitude smaller than the spontaneous emission time of the $b$ state as well as the typical time of heating loss, which are above 1 sec in this parameter regime [77].

With this we saw how to realize the outgoing Lindbladian (7). If desired one may also similarly realize the incoming Lindbladian (8) utilizing a reservoir of atoms in a third state $c$, as described in Sec. 3. The added complexity with respect to the equilibrium case is offset by the efficiency of the resulting relaxation process: While cooling fermionic atoms to low temperatures so as to approach an equilibrium ground state is generally hard [33,34], here the relaxation is engineered to bring the system to the desired state at a finite rate $\geq \gamma^{\text{in}}$, independent of the system size.

To demonstrate the utility of this approach I plot in Fig. 3(b) the resulting spectrum of the single-particle steady-state density matrix (9) [the occupancies $n_\lambda^{ss}(\mathbf{k})$], which deviate from the ideal values of 0 and 1 by less than 4%. The only exceptions are the edge modes, whose occupancies vary continuously between 0 to 1. This is a consequence of the steady state being topologically nontrivial, as discussed in the next Section. The behavior is qualitatively similar to the equilibrium case at small but finite temperature, though with a non-Fermi-Dirac distribution (with respect to the reference Hamiltonian), cf. Eq. (9). Employing the Kapit-Mueller reference Hamiltonian [70] truncated to a finite range may allow for arbitrarily low values of the flatness ratio. For example, adding to the Hofstadter reference Hamiltonian (13) the small next-nearest-neighbor terms of the Kapit-Mueller Hamiltonian results in steady state populations deviating by less than 0.7% from their ideal values, as depicted in Fig. 3(c).

---

[12]Note that this is of the order of $J_z$, so the separation of time scales between the system and bath dynamics assumed in the derivation of the Lindblad equation is not strictly obeyed. However, this Markovian limit is not actually necessary for the evaporative dynamics to function properly, as discussed in Sec. 3. Due to heating time limitations, in practice it will therefore be better to choose the indicated value of $\gamma^{\text{out}}$.

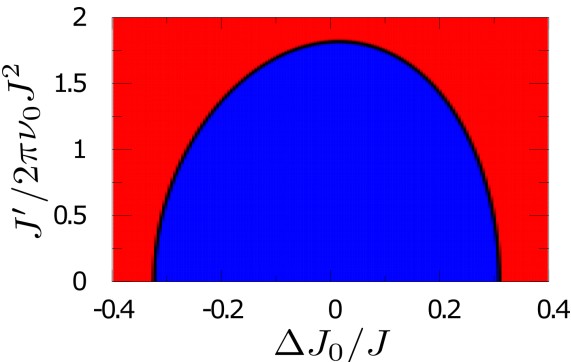

Figure 5: Phase diagram of the dissipative Hofstadter state as function of $\Delta J_0$, a shift in the onsite energy in Eq. (13), and of $J'$, the amplitude of a parasitic hopping Hamiltonian in the master equation (1). The blue (red) regions correspond to non-trivial (trivial) Chern number $C_1 = 1$ ($C_1 = 0$), where the Chern number is defined in Eq. (19) below. The parameter values are the same as in Fig. 3.

One may now introduce different perturbations, and study the stability of the state against them. The effects of two of them are presented in Fig. 5: (i) A deviation, $\Delta J_0$, of the onsite parameter $J_0$ [cf. Eq. (13)] from its optimal value; (ii) Finite Hamiltonian in Eq. (1), corresponding to residual nearest-neighbor tunneling with matrix element $J'$. As expected, the finite minimal decay rate ($\geq \gamma^{\text{in}}$) of all the modes under the combined Lindbladian (7)–(8) stabilizes the topological state against perturbations, so long as they do not reach the order of magnitude of the unperturbed couplings.

## 6 Topological classification and detection out of equilibrium

### 6.1 Topological mixed-state classification

In what sense should the mixed state produced by the above procedure be considered topological? To answer this question I will mostly follow the approach put forward in Refs. [7–9,13,52], though using a somewhat different argument. Topological classification of the ground states of quadratic fermionic Hamiltonians [28–31] is based on spectrally flattening the first quantized Hamiltonian, i.e., on replacing the eigenenergies by their signs [with respect to the chemical potential $\mu_0 = (\varepsilon_{1,\max} + \varepsilon_{2,\min})/2$; see Fig. 2(c)] and defining

$$Q(\mathbf{k}) = \sum_\lambda \text{sgn}[\varepsilon_\lambda(\mathbf{k}) - \mu_0] |\mathbf{k}\lambda\rangle \langle \mathbf{k}\lambda| . \tag{17}$$

This is a (first-quantization) operator that assigns the value $-1$ to all the occupied states, and $+1$ to all the unoccupied ones, and thus obeys $Q^2(\mathbf{k}) = 1$. Topological classification then proceeds as the study of the structure of the space of all possible such operators $Q(\mathbf{k})$.

Here comes the crucial observation: In equilibrium at zero temperature the matrix representing $Q(\mathbf{k})$ is simply related to the equilibrium single-particle reduced density matrix, $Q_{\mu\mu'}(\mathbf{k}) = 1 - 2\text{Tr}(\rho_{ss} a_{\mathbf{k}\mu}^\dagger a_{\mathbf{k}\mu'}) = 1 - 2G_{\mu\mu'}^{ss}(\mathbf{k})$. Out of equilibrium, for a quadratic Lindbladian of the type considered here, the steady state is still characterized by the steady state single-particle reduced density matrix $G_{\mu\mu'}^{ss}(\mathbf{k})$, but its eigenvalues (the occupancies) are not strictly 0 or 1 [cf. Fig. 3(b)–(c)]. However, as long as there is a gap between the occupancies which are closer to 0 and those which are closer to 1 (akin to the "purity gap" discussed in

Refs. [7–9, 13, 52][13]), one can extend the definition of $Q(\mathbf{k})$ to this case as

$$Q(\mathbf{k}) = \sum_{\lambda} \mathrm{sgn}\left[n_0 - n_\lambda^{ss}(\mathbf{k})\right] |\mathbf{k}\lambda\rangle \langle \mathbf{k}\lambda|, \tag{18}$$

where $n_0$ can take any value in the purity gap (similarly to the chemical potential in equilibrium zero-temperature systems), e.g., $n_0 = 1/2$, so that it is again amenable to topological classification by the procedures of Refs. [28–31]. Another way to formulate this approach is by defining the "entanglement Hamiltonian" between the system and the baths, $H^{\mathrm{ref}} \equiv -\log \rho_{ss}$. In the current case it is quadratic in the fermionic operators, with the same eigenmodes $|\mathbf{k}\lambda\rangle$ and with eigenvalues $\log\{[1 - n_\lambda^{ss}(\mathbf{k})]/n_\lambda^{ss}(\mathbf{k})\}$. As long as it is gapped, one may employ it to define the usual topological indices through its flattened version, which is exactly $Q(\mathbf{k})$ given by Eq. (18). For paired states one would need to include anomalous averages (of two creation or two annihilation operators) in the definition of $Q_{\mu\mu'}(\mathbf{k})$ as well. Let us note that in the presence of disorder the quasi-momentum $\mathbf{k}$ is not well defined, but one may introduce instead periodic boundary conditions with phases $\phi_x, \phi_y, \ldots$ in the different spatial directions, and classify the dependence of $Q$ on these phases [26, 78].

This approach has the pleasant property of assigning to equilibrium noninteracting systems at finite temperature the same sharply-quantized topological index as at zero temperature, reflecting the fact that the protected edge modes of such systems are not modified as function of temperature (although response functions do change, of course). This should be contrasted with other recent proposals, such as treating a non-pure steady state as a mixture of pure states and averaging over the topological index of these states, which does not lead to a quantized value [48]. Another recent approach advocates characterizing the density matrix in terms of the Uhlmann phase, which is quantized but shows spurious abrupt changes at a particular finite temperature in equilibrium [49,50]. The topological indexes defined in this work are not only free from these problems, but, as I will show in the next subsection, are experimentally measurable in cold atom systems.

Continuing with our dissipative Hofstadter example, one may now calculate the integer topological index, the first Chern number [79],

$$C_1 = \frac{1}{16\pi} \int \mathrm{Tr}\left\{Q(\mathbf{k})\left[\partial_{k_x} Q(\mathbf{k}), \partial_{k_y} Q(\mathbf{k})\right]\right\} \mathrm{d}\mathbf{k}, \tag{19}$$

where the integration is over the first Brillouin zone. This gives the value 1 for the states depicted in Fig. 3(b)–(c), which implies that there are edge eigenmodes of $G^{ss}$ whose occupancies must straddle the gap in the spectrum of $G^{ss}$, as indeed seen in Fig. 3(b)–(c). Let me also note that Eq. (10) shows that starting from, e.g., the trivial completely occupied [$n_\lambda(\mathbf{k}) = 1$ for all $\lambda$ and $\mathbf{k}$] or completely empty [$n_\lambda(\mathbf{k}) = 0$ for all $\lambda$ and $\mathbf{k}$] states, a gap in the spectrum of the occupancies will open up at arbitrarily short times, due to the gap in the spectrum of $\gamma^{\mathrm{out}}$. Thus, a nonzero Chern number and edge modes in the occupancy gap will immediately appear.

It should be noted that these edge modes are not decoupled modes of the dynamics. On the contrary, by Eq. (12) each mode in the system is damped by a rate $\gamma_\lambda^{\mathrm{out}}(\mathbf{k}) + \gamma^{\mathrm{in}} \geq \gamma^{\mathrm{in}}$, as explained above, so any deviation from the steady state, either in the bulk or at the edge, would decay at a finite rate, independent of the system size. As a matter of fact, since by Eq. (7) $\gamma_\lambda^{\mathrm{out}}(\mathbf{k}) = 2\pi\nu_0[\varepsilon_\lambda^{\mathrm{ref}}(\mathbf{k})]^2$, the decay rates of the edge modes span the gap between the decay rates of the almost-filled ($\lambda = 1$) and almost-empty ($\lambda \geq 2$) bands.

---

[13]As noted in these works, in contrast with equilibrium zero-temperature systems, the purity gap might close by varying the Lindbldian continuously without closing the gap of of the Lindbldian. For example, in our system one may increase $\gamma^{\mathrm{in}}$ to arbitrarily large values, and thus actually increase the gap of the Lindbldian, while driving the system towards the trivial completely occupied state, $n_\lambda^{ss}(\mathbf{k}) = 1$.

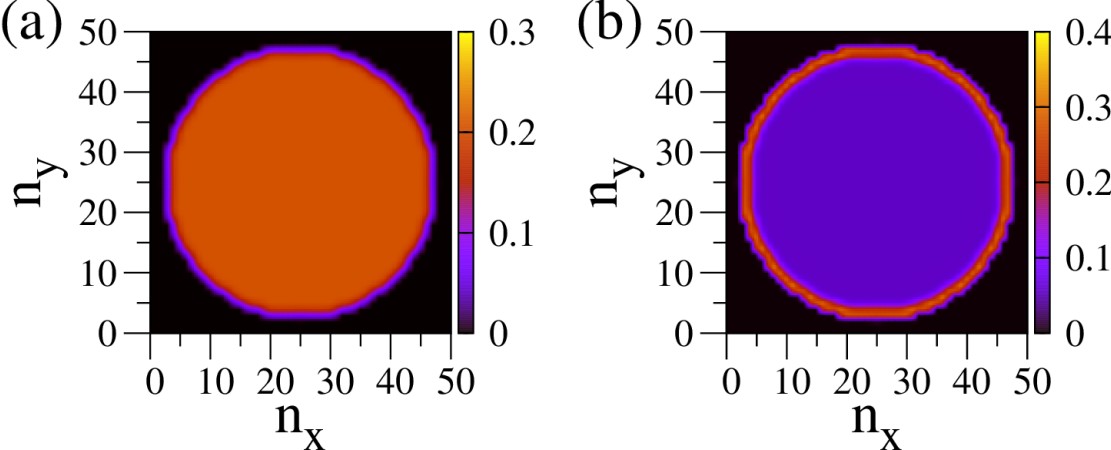

Figure 6: (a) A color map of the real-space particle distribution of the dissipative Hofstadter model on a $50 \times 50$ lattice with in a circular trap [simulated by adding a large state-independent evaporation rate outside the trap to $\mathcal{L}^{\text{out}}$, Eq. (7)]. Inside the trap the density $\approx 1/q = 1/5$ atoms per unit site, since only the lowest band of the reference Hamiltonian (13) is filled. The parameter values are as in Fig. 3. (b) A color map of the derivative of the local particle number with respect to $\gamma^{\text{in}}/(2\pi \nu_0 J^2)$. It is significant only near the edge, thus revealing the edge modes from Fig. 3.

One can shed further light on these results from a different perspective. There are different ways to define topological order. Some of them depend on the existence of anyons, manifested by topological degeneracy and entanglement entropy, and thus exclude integer quantum Hall or Chern insulators [67]. Others define a topologically-ordered state as one that cannot be created, starting from some trivial product state, by a finite depth local quantum circuit, or, equivalently, finite time (not scaling with the system size) evolution with a local Hamiltonian [80]. The last definition is usually taken to include Chern insulators, due to their gapless edge modes, which thus have long-range (power-law) correlations that cannot be created at a finite time by a local Hamiltonian. In our system the dynamics is dissipative but still local, hence one would expect similar scaling of the relaxation time to reach a topologically-nontrivial state. Indeed, such an approach has recently been suggested for the definition of topological order out of equilibrium [56]. Yet we found that pure Chern insulator states can be dissipatively created exponentially fast in time at a finite rate (finite gap of the Lindbladian) by a local (though not finite-range) Lindbaldian, which seems to be at odds with this argument. The resolution comes from the topological edge modes of $G^{ss}$. Their existence implies that one can only achieve the desired pure state in the bulk of the system. On the edge there must be edge modes of the density matrix whose average occupancies span the gap between 0 and 1 [Fig. 3(b)-(c)]. In a qualitatively similar manner to a finite temperature equilibrium state, this causes the edge correlations in the steady state of our system to decay exponentially with distance, so indeed there is no issue with creating these correlations at a finite rate by local Lindblad dynamics. In particular, this shows that while local Hamiltonian evolution of a pure state cannot to change its Chern number [81–84], local Lindblad evolution can, implying that Chern insulators, and actually all noninteracting topological states in any dimension, are not topological by the definition of Ref. [56].

## 6.2   Detection of nonequilibrium topology

How could the topologically-nontrivial states be detected experimentally? Here too one may employ many of the techniques suggested in equilibrium [35–44]. I will examine some of them concentrating on the dissipative Hofstadter state as an example. It should be noted that, differently from equilibrium, here the edge modes do not propagate but rather display pure decay, due to the purely-dissipative dynamics. Since there is no Hamiltonian term in the master equation (1), the conductivity cannot be straightforwardly defined. Hence, other indicators are needed.

One approach is to probe the real-space distribution of the atoms [85]. As shown in Fig. 6(a), it attains the constant value of $1/q$ per lattice site (corresponding to the lowest band, $\lambda = 1$, of the reference Hamiltonian (13) being almost filled, and all the others, $\lambda \geq 2$, being almost empty) in the bulk of the system, in analogy with the incompressibility of the corresponding equilibrium state. The partially-filled edge states [see Fig. 3(b)–(c)] could be revealed by taking the derivative of the density map with respect to the filling rate $\gamma^{\text{in}}$, which is significant only near the edge [Fig. 6(b)]. However, this is not an unambiguous indicator of a topologically-nontrivial state.

Different information can be revealed by examining the quasi-momentum distribution function [Fig. 7(a)], which can be inferred from a time-of-flight measurement, i.e., releasing the atoms and taking images of the expanding cloud [33,34]. The $q$-fold structure in the $k_x$ direction reflects the periodicity of the Hofstadter reference Hamiltonian (13), which is hidden in the real-space distribution, Fig. 6. To find the Chern number itself, Eq. (19), one may take a hybrid time-of-flight image [86] (see also [87]), in which the atoms are released in the $x$ direction but are kept confined in the $y$ direction, leading to a hybrid real space-momentum space distribution [Fig. 7(b)]: A nonzero Chern number, $C_1 \neq 0$, implies that the spatial density profile in the $y$ direction winds by $C_1$ unit cells of the reference Hamiltonian (13) ($C_1 q$ lattice sites) as $k_x$ winds through the Brillouin zone. Thus, the mixed state Chern number defined above is sharply-defined and measurable in an experiment. Let me note (following the discussion in the previous subsection of mixed-state topological classification) that for an equilibrium system at finite temperature this mixed-state Chern number will retain its value all the way up to infinite temperature, as only then would the contrast between the maxima and minima in Fig. 7(b) disappear.

# 7   Conclusions

To conclude, this work exposes the capabilities and limitations of dissipative preparation of topological states using quadratic Lindblad dynamics. We have seen that while finite-range gapped quadratic Lindbladians cannot lead to an exponential decay in time at a finite (system size independent) rate towards a unique pure steady state with nonzero Chern number, they can lead to a mixed state arbitrarily close to the desired pure one, which could be realized with cold atoms using currently available experimental techniques. The pure limit may be achieved if exponentially-local Lindbladians are allowed. I have also discussed the topological classification of such states in the mixed case, using the single-particle reduced density matrix, and the implications of topological nontriviality, such as the edge modes of the single-particle density matrix and the winding number of mixed position-momentum density maps.

While this study concentrated on Chern insulators, that is, 2D systems in symmetry class A, the extension to other topological classes [25–27] is clear. In particular, as mention in Sec. 4, the no-go theorem holds for these as well, following the no-go theorem for finite-range Hamiltonians with flat bands in topologically nontrivial states in two and higher dimensions [66]. One may of course argue that in the dissipative context, the limitations imposed by, e.g., time-

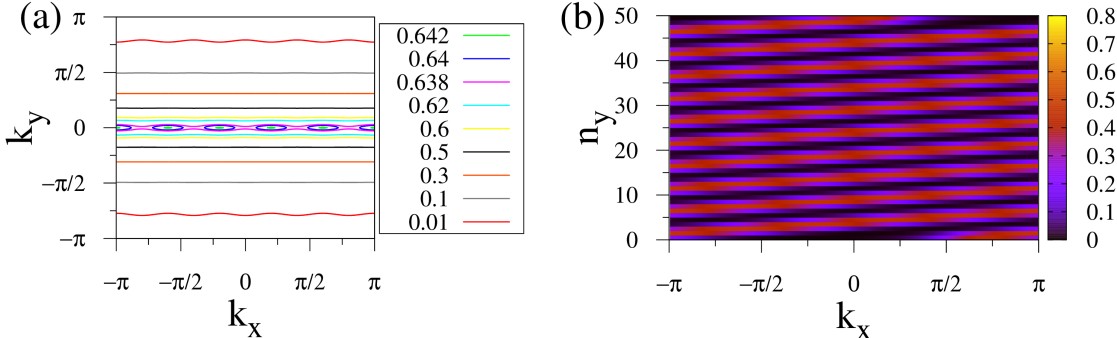

Figure 7: (a) A contour plot of the "quasi-momentum" (Fourier transform defined, here only, with respect to a $1 \times 1$, rather than $1 \times q = 1 \times 5$, unit cell) distribution of the dissipative Hofstadter model on the first Brillouin zone, which can be extracted from a time-of-flight measurement [33, 34]. The $q = 5$ fold periodicity in the $k_x$ direction reflects the periodicity of the Hofstadter reference Hamiltonian (13). The parameter values are as in Fig. 3. (b) A color map of the mixed quasi momentum-real space distribution (along the $x$ and $y$ directions, respectively), which can be inferred from a hybrid time-of-flight measurement [86]. The spatial density profile in the $y$ direction winds by one unit cell (of size $q = 5$) as $k_x$ winds through the Brillouin zone, showing that $C_1 = 1$ [cf. Eq. (19)]. The system is a 50-site-wide strip whose long axis is $x$; similar results hold in the presence of a trap [86]. $k_x$ and $k_y$ are here dimensionless, or, equivalently, measured in units of $1/d$, the inverse 2D lattice spacing.

reversal symmetry, lose their physical significance. If these are relaxed, pure steady states can result from finite-range gapped Lindbladian. This work was centered at cold atom systems, but the ideas introduced here should find applications to other types of "quantum simulators", including, among others, superconducting nanocircuits, quantum dots, trapped ions, and nuclear and impurity spins [32].

These results give rise to many important questions for future study. Among them is the effects of including the Hamiltonian in the Lindblad dynamics (1). While this neither modifies the results of the no-go theorem nor is necessary for our recipe as we have seen above, it may allow one to define currents and hence, e.g., the Hall conductivity. Our recent results [88] indicate that the latter is not generally quantized in the driven-dissipative case. A similar conclusion holds for Laughlin-type charge pumping [89]. Thus, the equilibrium relation between topology, edge modes, and quantized responses does not generalize unmodified to nonequilibrium situations, an interesting topic for further research.

Even more exciting directions open up when going beyond quadratic dynamics, thus allowing for the creation of strongly-correlated dissipative states. In the topological context, states which were distinct in the noninteracting case may become topologically-equivalent (first realized in 1D [90, 91]), while new phases could arise [19, 58–62, 67, 80, 92, 93]. In particular, above 1D the inclusion of interactions may lead to "topologically-ordered states" (e.g., fractional quantum Hall), whose excitations display fractional charge and statistics (anyons). The statistics could even be nonabelian, raising the possibility of topological quantum computation [94]. It would be interesting to understand the dissipative analogues of these, and the even more exciting possibility of novel topological phenomena which occur only out of equilibrium.

# Acknowledgements

I would like to thank B. A. Bernevig, B. Bradlyn, J. Budich, J. I. Cirac, N. Cooper, S. Diehl, M. Hafezi, Y. Gefen, A. Gorshkov, A. Kemenev, N. Schuch, H. H. Tu, and P. Zoller for useful discussions.

**Funding information**   This research was supported by the Israel Science Foundation (Grant No. 227/15), the German Israeli Foundation (Grant No. I-1259-303.10), the US-Israel Binational Science Foundation (Grant No. 2016224), and the Israel Ministry of Science and Technology (Contract No. 3-12419).

# A   Derivation of the general quadratic fermionic Lindblad equation

Let us consider a general noninteracting model of fermions composed of a system $S$ and an environment $E$. The latter could represent a combination of several independent baths. We assume that the total number of fermions is conserved, though fermions may be exchanged between the system and the environment. Thus, one may write:

$$H = H_S + H_E + H_{SE}, \tag{20}$$

where

$$H_S = \sum_{m,m'} h_{mm'}^S a_m^\dagger a_{m'} = \sum_p \varepsilon_p^S a_p^\dagger a_p, \tag{21}$$

$$H_E = \sum_{n,n'} h_{nn'}^E b_n^\dagger b_{n'} = \sum_q \varepsilon_q^E b_q^\dagger b_q, \tag{22}$$

$$H_{SE} = \sum_{m,n} h_{mn}^{SE} a_m^\dagger b_n + \text{h.c.} = \sum_{p,q} h_{pq}^{SE} a_p^\dagger b_q + \text{h.c.} \tag{23}$$

Here $h_{mm'}^S$ is a Hermitian matrix representing the system single-particle Hamiltonian in some basis (e.g., localized orbitals in real space) and $\varepsilon_p^S$ its eigenvalues. The fermionic annihilation operators in the corresponding bases ($a_m$ and $a_p$, respectively) are related by a unitary transformation $u^S$ as $a_p = \sum_m u_{pm}^S a_m$, where $h_{mm'}^S = \sum_p \varepsilon_p^S (u_{pm}^S)^* u_{pm'}^S$. Similarly, $h_{nn'}^E$ is a Hermitian matrix representing the environment single-particle Hamiltonian in some basis (e.g., localized orbitals in real space) and $\varepsilon_q^E$ its eigenvalues. The fermionic annihilation operators in the corresponding bases ($b_n$ and $b_q$, respectively) are related by a unitary transformation $u^E$ as $b_q = \sum_n u_{qn}^E b_n$, where $h_{nn'}^E = \sum_q \varepsilon_q^E (u_{qn}^E)^* u_{qn'}^E$. Finally, the system-bath coupling matrices in the two bases are related by $h_{mn}^{SE} = \sum_{pq} h_{pq}^{SE} (u_{pm}^S)^* u_{qn}^E$. We may now go to the interaction representation with respect to the system-bath coupling, eliminating $H_S + H_B$ in favor of making $H_{SB}$ time dependent, $H_{SE}^I = \sum_{m,n} h_{pq}^{SE} e^{i(\varepsilon_p^S - \varepsilon_q^B)t} a_p^\dagger b_q + \text{h.c.}$

Now one may proceed along the standard derivation of the Lindblad master equation [2], which will be reproduced here for completeness. I will denote the density matrix of the system+environment in the interaction representation by $\rho_{SE}^I$. It obeys $\partial_t \rho_{SE} = -i[H_{SE}^I, \rho_{SE}^I]$. Integrating this equation from an initial time $t_0$ to time $t$, substituting the result into the original equation, and then taking the trace over the environment, one finds the following equation for the system density matrix in the interaction representation, $\rho_S^I = \text{Tr}_E \rho_{SE}^I$:

$$\partial_t \rho_S^I(t) = -i\text{Tr}_E\left[H_{SE}^I(t), \rho_{SE}^I(t_0)\right] - \int_{t_0}^t \mathrm{d}t' \text{Tr}_E\left[H_{SE}^I(t), \left[H_{SE}^I(t'), \rho_{SE}^I(t')\right]\right]. \tag{24}$$

Assuming that at $t_0$ the system and the environment are uncorrelated, $\rho_{SE}^I(t_0) = \rho_S^I(t_0) \otimes \rho_E^I(t_0)$, the first term on the right hand side of the last equation vanishes, since it reduces to a sum over product of averages of single fermion operators over the system and environment initial states. As for the second term, it is second order in the system-environment coupling. Thus, to the lowest Born approximation one may take a factorized form for the system+environment density matrix, and further assume the environment state is approximately constant in time (i.e., the baths are "large", hence not affected by the system), $\rho_{SE}^I(t) \approx \rho_S^I(t) \otimes \rho_E^I$. Moreover, one may make the Markov approximation, namely assume that the bath dynamics is much faster than that of the system, so for the values of $t'$ which significantly contribute in the last equation one may approximate $\rho_S^I(t') \approx \rho_S^I(t)$. This also allows to take the lower integral limit to $-\infty$. We can finally write

$$\partial_t \rho_S^I(t) = -\int_0^\infty d\tau \, \mathrm{Tr}_E \left[ H_{SE}^I(t), \left[ H_{SE}^I(t-\tau), \rho_S^I(t) \otimes \rho_E^I \right] \right]. \tag{25}$$

The right hand side then breaks into a sum of averages of two environment operators over the environment state, times a term containing two system operators.

Let us now specialize to our quadratic fermionic system:

$$\partial_t \rho_S^I(t) = \sum_{p,p',q,q'} \left( h_{pq}^{SE} \right)^* h_{p'q'}^{SE} \int_0^\infty d\tau \, e^{-i(\varepsilon_p^S - \varepsilon_q^E)t + i(\varepsilon_{p'}^S - \varepsilon_{q'}^E)(t-\tau)}$$
$$\left\{ \mathrm{Tr}_E \left( \rho_E^I b_{q'} b_q^\dagger \right) \left[ a_p \rho_S^I(t) a_{p'}^\dagger - \rho_S^I(t) a_{p'}^\dagger a_p \right] \right.$$
$$\left. \mathrm{Tr}_E \left( \rho_E^I b_q^\dagger b_{q'} \right) \left[ a_{p'}^\dagger \rho_S^I(t) a_p - a_p a_{p'}^\dagger \rho_S^I(t) \right] \right\}$$
$$+ \text{h.c.} \tag{26}$$

Let us assume that $\mathrm{Tr}_E(\rho_E^I b_{q'} b_q^\dagger)$ and $\mathrm{Tr}_E(\rho_E^I b_q^\dagger b_{q'})$ do not vanish only for $q = q'$. Furthermore, let us make the usual assumption that non-degenerate system states have much larger spacing with respect to the relaxation rate (which we are about to write down explicitly), hence one can make the rotating-wave approximation and keep only terms with $p = p'$ in Eq. (26). With that one obtains:

$$\partial_t \rho_S^I(t) = \sum_p \Gamma_p^{\text{out}} \left[ 2 a_p \rho_S^I(t) a_{p'}^\dagger - a_{p'}^\dagger a_p \rho_S^I(t) - \rho_S^I(t) a_{p'}^\dagger a_p \right]$$
$$+ \Gamma_p^{\text{in}} \left[ 2 a_p^\dagger \rho_S^I(t) a_{p'} - a_{p'} a_p^\dagger \rho_S^I(t) - \rho_S^I(t) a_{p'} a_p^\dagger \right], \tag{27}$$

where the coefficients $\Gamma_p^{\text{out}}$, $\Gamma_p^{\text{in}}$ are given by

$$\Gamma_p^{\text{out}} = \sum_q \left| h_{pq}^{SE} \right|^2 \int_0^\infty d\tau \, e^{-i(\varepsilon_p^S - \varepsilon_q^E)\tau - \eta\tau} \mathrm{Tr}_E \left( \rho_E^I b_q b_q^\dagger \right)$$
$$= \sum_q \left| h_{pq}^{SE} \right|^2 \mathrm{Tr}_E \left( \rho_E^I b_q b_q^\dagger \right) \left[ \pi \delta \left( \varepsilon_q^E - \varepsilon_p^S \right) + i\mathcal{P} \frac{1}{\varepsilon_q^E - \varepsilon_p^S} \right], \tag{28}$$

$$\Gamma_p^{\text{in}} = \sum_q \left| h_{pq}^{SE} \right|^2 \int_0^\infty d\tau \, e^{i(\varepsilon_p^S - \varepsilon_q^E)\tau - \eta\tau} \mathrm{Tr}_E \left( \rho_E^I b_q^\dagger b_q \right)$$
$$= \sum_q \left| h_{pq}^{SE} \right|^2 \mathrm{Tr}_E \left( \rho_E^I b_q^\dagger b_q \right) \left[ \pi \delta \left( \varepsilon_q^E - \varepsilon_p^S \right) - i\mathcal{P} \frac{1}{\varepsilon_q^E - \varepsilon_p^S} \right], \tag{29}$$

where $\eta \to 0^+$ is a convergence factor, and $\mathcal{P}$ denotes Cauchy's principal value. Twice the real parts of these coefficients, $\gamma_p^{\text{out}} \equiv 2\mathrm{Re}(\Gamma_p^{\text{out}})$ and $\gamma_p^{\text{in}} \equiv 2\mathrm{Re}(\Gamma_p^{\text{in}})$, are the dissipation rates (rates



of particles leaving the system into the bath or entering the system from the bath, respectively) as given by the Fermi golden rule. The imaginary parts, $\Delta\varepsilon_p^{\text{out}} \equiv \text{Im}(\Gamma_p^{\text{out}})$ and $\Delta\varepsilon_p^{\text{in}} \equiv -\text{Im}(\Gamma_p^{\text{in}})$, are corrections to the system eigenenergies ("Lamb shifts"). Returning to the Schrodinger picture and to the original basis, we obtain the fermionic Lindblad master equation for the system density matrix $\rho_S$ (which is denoted by $\rho$ in the rest of the paper), Eq. (2), with $\tilde{h}_{mm'}^S = \sum_p (\varepsilon_p^S + \Delta\varepsilon_p^{\text{out}} + \Delta\varepsilon_p^{\text{in}})(u_{pm}^S)^* u_{pm'}^S$ being the Hermitian system Hamiltonian modified by the Lamb shift corrections, while $\gamma_{mm'}^{\text{out}} = \sum_p \gamma_p^{\text{out}} u_{pm}^S (u_{pm'}^S)*$, $\gamma_{mm'}^{\text{in}} = \sum_p \gamma_p^{\text{in}} (u_{pm}^S)^* u_{pm'}^S$ are Hermitian positive semi-definite decay rate matrices.

As noted in Sec. 2 above, this derivation can be extended to the case when fermion number in the system+environment is not conserved due to the presence of pairing terms ($a_m a_{m'}$, $a_m b_n$, $b_n b_{n'}$ and their conjugates) in the total Hamiltonian (20). The end result would be (beyond modifications to the expressions for the previously-defined coefficients) that the Hamiltonian part of the Lindblad equation would be supplemented by new terms of the form $\Delta_{mm'} a_m a_{m'} + \text{h.c.}$, and similarly, the Lindbladian would include new terms of the form $\gamma_{mm'}^{\text{pair}}[2a_m \rho_S a_{m'} - a_{m'} a_m \rho_S - \rho_S a_{m'} a_m]/2 + \text{h.c.}$, with antisymmetric matrices $\Delta_{mm'}$ and $\gamma_{mm'}^{\text{pair}}$.

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
