# Peer review of "Dissipation-induced topological insulators: A no-go theorem and a recipe"

_SciPost Physics, doi:SciPost Phys. 7, 067 (2019)_

## Round 1 · Referee Report · Anonymous (Referee 1) · 2019-8-28

Report

In this manuscript, the author have discussed a general framework to generate 2D Chern insulators from quadratic Lindblad dynamics. To be specific, the author have introduced a recipe to generate the reference Hamiltonian (as the parent Hamiltonian for the target Chern insulator state) by coupling the target 2D fermionic system to two different baths. The author then proved a no-go theorem, and showed that the recipe can generate either i) a topological pure state by exponentially-local Lindbladian, or ii) a mixed state that is close to the target pure state by a finite-range Lindbladian. The author use the dissipative Hofstadter model as a concrete example to realise the recipe. The author then discussed the way to define Chern number for this dissipative case, as well as the schemes to detect the nonequilibrium topology by in-situ density image or by hybrid TOF image.

The generation of topological states from dissipation provides an important alternative compared with pure coherent quantum dynamics. Certain previous studies have focused on the dissipative generation of topological superconductors. This work makes the dissipative generation of topological insulators possible. I think it deserves publication in SciPost Physics. However, before my recommendation, the following points (see Requested changes) should be considered appropriately.

Requested changes

  1. The scope of the “no-go theorem” is not clear enough. In Sec. 4 titled “No-go theorem”, the explicit statement of the theorem could not be found. The author says that “I will now proceed to prove that one cannot do better than the above recipe, that is, one cannot obey all the demands posed in Sec. 2, and achieve exponential decay at a finite rate towards a pure dark state with nonzero Chern number in 2D out of finite-range Lindblad dynamics.” In my limited understanding, this is not the suitable way to express a theorem. I think a theorem should be self-contained (without refereeing to certain indefinite contents like “the above recipe”, “the demands posed in Sec. 2”, etc.) and concise. It should include all the definite preconditions, and then be stated clearly. In addition, it seems to me that the theorem here just applies to the dissipative processes discussed in Sec. 3, i.e., the case that the system is embedded in two baths with decay rates $\gamma^\rm{in}$ and $\gamma^\rm{out}$. But in principle, the author could not eliminate other possibilities of dissipatively generating Chern insulators. 

  2. In Sec. 5, the author give a concrete example to realise the Chern insulator state using the author’s recipe. It is not clear to the readers how the scheme in Fig. 4 can generate the Lindbladian Eq. (6), and further how the corresponding Lindbladian take the form of the Hofstadter Hamiltonian Eq. (11). I would suggest the author write down the explicit coupling Hamiltonian for the scheme in Fig. 4 and make a detailed derivation on how to reach Eq. (11) from this scheme. In particular, the author should show how the decay rates $\gamma^\rm{in}$ and $\gamma^\rm{out}$ are related to the parameters in Fig. 4. 

  3. In Fig. 5, the author show the Chern number as the phase diagram of the dissipative Hofstadter state for different parameters. As long as the Chern number can be defined from a time-dependent state, could the author also show the build-up process of the Chern number as a function of time from the Lindbladian dynamics? 

  4. The author says that one can construct “a mixed state arbitrarily close to the desired pure one via finite-range dynamics”. However, the term “close” should be quantified. For instance, if we define the fidelity between the target pure state and the mixed state prepared by the author’s recipe, how does the fidelity converge to one as a function of the parameters in the author’s recipe? 

  5. There are a few typos in the manuscript. For instance, the title of Sec. 5, “Example: Dissipative Hofstadter with cold atoms”, it should be “Example: Dissipative Hofstadter model with cold atoms”. Likewise, “(intereger quantum Hall on a lattice)” on page 2, should be “(integer quantum Hall states on a lattice)”.

  • validity: good
  • significance: high
  • originality: high
  • clarity: good
  • formatting: good
  • grammar: good

Author:  Moshe Goldstein  on 2019-10-30  [id 636]

(in reply to Report 1 on 2019-08-28)

I would like to thank the Referee for carefully reading the manuscript. I am glad that the Referee found the work interesting and appropriate for publication in SciPost. The Referee’s comments, for which I am thankful and which I have fully implemented, mainly concern the presentation, and helped me in improving it, as detailed below.

1) The Referee wrote: ``The scope of the “no-go theorem” is not clear enough. In Sec. 4 titled “No-go theorem”, the explicit statement of the theorem could not be found. The author says that “I will now proceed to prove that one cannot do better than the above recipe, that is, one cannot obey all the demands posed in Sec. 2, and achieve exponential decay at a finite rate towards a pure dark state with nonzero Chern number in 2D out of finite-range Lindblad dynamics.” In my limited understanding, this is not the suitable way to express a theorem. I think a theorem should be self-contained (without refereeing to certain indefinite contents like “the above recipe”, “the demands posed in Sec. 2”, etc.) and concise. It should include all the definite preconditions, and then be stated clearly. In addition, it seems to me that the theorem here just applies to the dissipative processes discussed in Sec. 3, i.e., the case that the system is embedded in two baths with decay rates \gamma^{in} and \gamma^{out}. But in principle, the author could not eliminate other possibilities of dissipatively generating Chern insulators.’’

Reply: Following the Referee’s comment, as well as a similar comment by the other Referee, I have formulated the central results of the work as a theorem in Section 2, and also revised some of the definitions appearing there to make them clearer and more precise. The abstract was also expanded to make it clearer. In addition, following a comment by the other Referee, an Appendix was added which details the derivation of the most general quadratic fermionic Lindblad equation, which is given in the text as the new Eq. (2). As the derivation shows, and is now stated explicitly in the text, even for an arbitrary number of baths which are coupled to the system via arbitrary couplings, the Lindblad equation will end up containing two terms, one with rate matrix \gamma^{in} and one with rate matrix \gamma^{out}. Each bath may in principle contribute to both of these terms. Thus, the no-go theorem is general to the extent stated in the text. For simplicity the explicit construction in Section 3 indeed involves two bath, one contributing to \gamma^{in} and one to \gamma^{out}, but, again, this is not a limitation on the no-go theorem. Let me also note that when one allows for ``pairing’’ terms, which do not conserve the total number of particles in the system + baths, additional terms appear in the Lindbald equation, but the theorem remains valid, as explained in the last paragraph of Section 4 (already in the previous version).

2) The Referee wrote: ``In Sec. 5, the author give a concrete example to realise the Chern insulator state using the author’s recipe. It is not clear to the readers how the scheme in Fig. 4 can generate the Lindbladian Eq. (6), and further how the corresponding Lindbladian take the form of the Hofstadter Hamiltonian Eq. (11). I would suggest the author write down the explicit coupling Hamiltonian for the scheme in Fig. 4 and make a detailed derivation on how to reach Eq. (11) from this scheme. In particular, the author should show how the decay rates $\gamma^{in}$ and $\gamma^{out}$ are related to the parameters in Fig. 4.’’

Reply: The discussion in Section 5 was extensively elaborated in line with the Referee’s comment. To make it a bit simpler I now make the specific choice that the b-atoms, while flying away, experience a 3D tetragonal lattice optical potential. Eq. (6) in Section 3 was therefore also modified accordingly.

3) The Referee wrote: ``In Fig. 5, the author show the Chern number as the phase diagram of the dissipative Hofstadter state for different parameters. As long as the Chern number can be defined from a time-dependent state, could the author also show the build-up process of the Chern number as a function of time from the Lindbladian dynamics?’’

Reply: As discussed in the paragraph surrounding Eq. (10) (a discussion which was somewhat expanded following the next comment), generic observable, such as the state occupancies, decay exponentially in time towards their steady state value. The Chern number is a bit peculiar in this respect, since it can only assume integer values, if $n_0$ is chosen in a purity gap (similarly to the chemical potential in equilibrium zero-temperature systems). For example, if one starts from either the completely occupied or completely empty state, the occupancies will decay towards their steady state values exponentially in time, and a gap in the occupancy spectrum will open up immediately. Thus, the Chern number associated with this gap will assume a nonzero integer value immediately, and, as a consequence, edge modes of the density matrix [similar to those depicted in Figs. 3(b)-(c)] will appear immediately, although for very short time the gap will be very small and the edge modes will be hard to detect. This is now explained in the paragraph surrounding Eq. (19).

4) The Referee wrote: ``The author says that one can construct “a mixed state arbitrarily close to the desired pure one via finite-range dynamics”. However, the term “close” should be quantified. For instance, if we define the fidelity between the target pure state and the mixed state prepared by the author’s recipe, how does the fidelity converge to one as a function of the parameters in the author’s recipe?’’

Reply: What was meant by this statement is that the fidelity approached unity exponentially with the range of the Lindbladian. This is now included in the statement of the theorem in Section 2, and is explained in more details in Section 3 in the paragraph surrounding Eq. (9). The discussion of convergence as function of time in the following paragraph was also clarified correspondingly.

5) The Referee wrote: ``There are a few typos in the manuscript. For instance, the title of Sec. 5, “Example: Dissipative Hofstadter with cold atoms”, it should be “Example: Dissipative Hofstadter model with cold atoms”. Likewise, “(intereger quantum Hall on a lattice)” on page 2, should be “(integer quantum Hall states on a lattice)”.’’

Reply: These typos, as well as some others, have been corrected.

---

## Round 1 · Referee Report · Anonymous (Referee 2) · 2019-9-4

Report

This work focuses on a class of gapped quadratic fermionic Lindblad master equations:

(1) First, a recipe is provided for Lindbladians whose steady states are ground states of arbitrary non-interacting fermionic tight-binding Hamiltonians. Each of the required fermionic modes has two additional baths, which are then traced out to obtain each mode's two personal dissipators. The coefficients of each dissipator are tuned so that the steady state of the sum of all dissipators is close to the ground state of the desired tight-binding Hamiltonian.

(2) Later, the main claim is proven, namely, that pure steady states of finite-range gapped quadratic fermionic Lindbladians have to be topologically trivial. The proof is by contradiction. It assumes finite range, steady-state purity, existence of a gap, and topological nontriviality. It then shows that there cannot be a gap -- the contradiction. A key step uses recent developments by J. Dubail and N. Read [64-65] dealing with fiber bundles over the Brillouin zone.

(3) A numerical example stabilizing bands of the Hofstadter model is provided, along with details tailored to realizing this example in cold atoms. A topological classification is provided, where the topological properties of a (possibly mixed) Gaussian fermionic state rho are the same as those of the pure state that is "closest" to rho in the distance imposed by eigenvalues of the correlation matrix. A discussion of how to detect said topology in experiments follows.

This is an interesting link between topological phases and open systems, particularly since the technical parts are viewed from the lens of hard condensed matter. I am not qualified to judge the accuracy of the proof. Physically however, helped by the useful discussions at the end of Sec. 6.1, it seems these results are not at odds with the notion that "true" topological order cannot be constructed using finite-time local evolution.

I recommend publishing this work eventually. However, it took three readings to start to distill the above points (1-2) of this work. I thus would prefer that the comments below are first addressed in order for this to be accessible to more than a small expert community.

Requested changes

  • More clarification regarding "uniqueness" of the steady state could help. Per Prosen (arXiv:0801.1257, Thm. 1), a quadratic fermionic Lindbladian has a unique steady state if none of the participating fermionic eigenmodes have zero eigenvalue. The first sentence of Sec. 3 claims to provide a recipe for such a Lindbladian. It seems this Lindbladian is a sum of Eqs. (6-7), and not Eq. (6) alone (which in fact has a zero mode if the lowest band is flat). This is something the author seems to be aware of in footnote 5 and at the very bottom of pg. 8, but in the discussion below Eq. (3), the state corresponding to the zero mode lambda = 1 is also deemed "unique". It may be useful to first state the family L = L_in + L_out explicitly, and then focus on what properties of L are required, respectively, for finite range, steady-state uniqueness, steady-state purity, existence of a gap, and topological nontriviality.

  • The tracing out of the bath seems like a fermionic extension of the "standard" protocol of adiabatic elimination, discussed e.g. in [arXiv:0803.1447, APPENDIX: ENGINEERING DISSIPATION] or in great detail in Carmichael [doi:10.1007/978-3-662-03875-8, Sec. 1.4]. It would be helpful to those not as well-acquainted with fermions to explicitly see the elimination of the modes b_{i,mu} in an Appendix.

  • "Polynomial bundles" from [64-65] are casually invoked on pg. 10. The context of those works is tensor network states, but here one deals with Gaussian fermionic states. More explanation of the link to [64-65] is warranted.

  • A claim is made that the topological classification stemming from Eq. (13) does not undergo "spurious abrupt changes at a particular ?finite temperature". I am not convinced of this yet, because it is not clear to me how this scheme applies to states whose occupancies are not close to 0 or 1. In this classification, is a steady state that is 99% in the lowest band (and 1% in the second lowest) equivalent to one that is 51% in the lowest band (and 49% in the second lowest)? Is it possible to have a continuous family L(t) such that, at t=0, they stabilize a state that is mostly in the lowest band, while at t=1, they stabilize a state most of which is split evenly between the two lowest bands?

  • The four assumptions in the no-go proof (finite range, steady-state purity, existence of a gap, and topological nontriviality) are somewhat scattered throughout Sec. 4. A more concise statement is warranted. It may be helpful to borrow from the mathematicians and use "Theorem" notation.

  • There are several quite long difficult-to-parse sentences. For example, the "key idea" at the bottom of page five basically describes the entire recipe in one long run-on sentence.

  • validity: good
  • significance: good
  • originality: good
  • clarity: good
  • formatting: reasonable
  • grammar: acceptable

Author:  Moshe Goldstein  on 2019-10-30  [id 635]

(in reply to Report 2 on 2019-09-04)
Category:
answer to question
correction

I would like to thank the Referee for carefully reading the manuscript. I am glad that the Referee found the work interesting and appropriate for publication in SciPost. The Referee’s comments, for which I am thankful and which I have fully implemented, mainly concern the presentation, and helped me in improving it, as detailed below.

1) The Referee wrote: ``More clarification regarding "uniqueness" of the steady state could help. Per Prosen (arXiv:0801.1257, Thm. 1), a quadratic fermionic Lindbladian has a unique steady state if none of the participating fermionic eigenmodes have zero eigenvalue. The first sentence of Sec. 3 claims to provide a recipe for such a Lindbladian. It seems this Lindbladian is a sum of Eqs. (6-7), and not Eq. (6) alone (which in fact has a zero mode if the lowest band is flat). This is something the author seems to be aware of in footnote 5 and at the very bottom of pg. 8, but in the discussion below Eq. (3), the state corresponding to the zero mode lambda = 1 is also deemed "unique". It may be useful to first state the family L = L_in + L_out explicitly, and then focus on what properties of L are required, respectively, for finite range, steady-state uniqueness, steady-state purity, existence of a gap, and topological nontriviality.’’

Reply: As indicated by the Referee, we are in complete agreement that both L_{out} and L_{in} are required for having a unique steady state. Still, I believe the current order of presentation allows to introduce the ideas, especially the role of the flat band, more gradually. The discussion in Section 3 then leads to Section 4, where the presentation is more in line with the approach advocated by the Referee. But to prevent any confusion on behalf of the reader in Section 3, I have now expanded the previous comment 5 mentioned by the Referee into a half of a paragraph, appearing at the end of the paragraph following the one surrounding Eq. (7), and also made sure the word ``unique’’ appears only when appropriate. Further clarifications are in a new paragraph at the end of Section 3, which replaces the previous comment 9. In addition, I now cite Prosen’s paper mentioned by the Referee (as the new Ref. [57]) in several locations where results he proves were previously taken for granted.

2) The Referee wrote: ``The tracing out of the bath seems like a fermionic extension of the "standard" protocol of adiabatic elimination, discussed e.g. in [arXiv:0803.1447, APPENDIX: ENGINEERING DISSIPATION] or in great detail in Carmichael [doi:10.1007/978-3-662-03875-8, Sec. 1.4]. It would be helpful to those not as well-acquainted with fermions to explicitly see the elimination of the modes b_{i,mu} in an Appendix.’’

Reply: An appendix was added with the details of the derivation of the quadratic fermionic Lindblad equation.

3) The Referee wrote: ``"Polynomial bundles" from [64-65] are casually invoked on pg. 10. The context of those works is tensor network states, but here one deals with Gaussian fermionic states. More explanation of the link to [64-65] is warranted.’’

Reply: The definition of polynomial bundles in Refs. [64-65] ([65-66] in the revised manuscript) is a set of bands whose wavefunctions solve a system of linear equations with coefficients which are trigonometric polynomials in k. The latter form results from having short range in the real space representation. These works indeed discuss the relation between tensor network states and polynomial bundles, but, again, the definition of polynomial bundles does not involve tensor networks. This definition therefore directly applies to the filled as well as to the empty bands in the discussion in Section 4. In the text I have elaborated the definition of polynomial bundles to make this clearer.

4) The Referee wrote: ``A claim is made that the topological classification stemming from Eq. (13) does not undergo "spurious abrupt changes at a particular finite temperature". I am not convinced of this yet, because it is not clear to me how this scheme applies to states whose occupancies are not close to 0 or 1. In this classification, is a steady state that is 99% in the lowest band (and 1% in the second lowest) equivalent to one that is 51% in the lowest band (and 49% in the second lowest)? Is it possible to have a continuous family L(t) such that, at t=0, they stabilize a state that is mostly in the lowest band, while at t=1, they stabilize a state most of which is split evenly between the two lowest bands?’’

Reply: As explained in the text, what is important is that there is a gap in the spectrum of the density matrix, that is, a gap in the spectrum of occupancies. As long as the gap exists, one may define, e.g., the Chern number (choosing n_0 inside the gap), and if it is nonzero, edge modes will appear in that gap in the spectrum of the density matrix, similar to those plotted in Fig. 3(b)-(c). These edge modes will remain there even if the occupancy ratio between the bands changes from 99:1 to 51:49. As the Referee wrote, and as has actually been noted by previous works cited in the text, one may close the purity gap and change the nonequilibrium topological index (and thus eliminate the edge modes of the density matrix) while keeping the gap of the Lindbladian finite. I added a new comment 13 to the paragraph surrounding Eq. (18) where this is stated, and clarified by a concrete example.

5) The Referee wrote: ``The four assumptions in the no-go proof (finite range, steady-state purity, existence of a gap, and topological nontriviality) are somewhat scattered throughout Sec. 4. A more concise statement is warranted. It may be helpful to borrow from the mathematicians and use "Theorem" notation.’’

Reply: Following the Referee’s comment, as well as a similar comment by the other Referee, I have formulated the central results of the work as a theorem in Section 2, and also revised some of the definitions appearing there to make them clearer and more precise.

6) The Referee wrote: ``There are several quite long difficult-to-parse sentences. For example, the "key idea" at the bottom of page five basically describes the entire recipe in one long run-on sentence.’’

Reply: The sentence indicated by the Referee was broken into three shorter ones. Similar modifications were made to other long sentences throughout the text.

---

## Round 4 · Referee Report · Anonymous (Referee 1) · 2019-11-4

Report

In this revised manuscript, the author has addressed appropriately all the points raised in my previous report, and I am now glad to recommend it for publication.

By the way, it would be better to split the newly added Eq. (15) into two lines.

---

## Round 4 · Author Response

Dear Editor,
I would like to thank both Referees for carefully reading the manuscript. I am glad that they both found the work interesting and appropriate for publication in SciPost. Their comments, for which I am thankful and which I have fully implemented, mainly concern the presentation, and helped me in improving it. In the reply to each referee I detail the specific changes made in response to each comment. With this I believe the manuscript is ready for publication.

Sincerely yours,
Moshe Goldstein

---

## Round 4 · List of Changes

The modifications to the manuscript are detailed in my responses to the Referees.

---

## Editorial Decision

published